# Macrophage spatial heterogeneity in gastric cancer defined by multiplex immunohistochemistry

Yu-Kuan Huang [1,2,3], Minyu Wang [1,2,3], Yu Sun [2,3], Natasha Di Costanzo[1], Catherine Mitchell[4], Adrian Achuthan [3], John A. Hamilton [3,5], Rita A. Busuttil [1,2,3] & Alex Boussioutas[1,2,3]

Tumor-associated macrophages (TAMs), one of the most abundant immune components in gastric cancer (GC), are difficult to characterize due to their heterogeneity. Multiple approaches have been used to elucidate the issue, however, due to the tissue-destructive nature of most of these methods, the spatial distribution of TAMs in situ remains unclear. Here we probe the relationship between tumor context and TAM heterogeneity by multiplex immunohistochemistry of 56 human GC cases. Using distinct expression marker profiles on TAMs, we report seven predominant populations distributed between tumor and non-tumor tissue. TAM population-associated gene signatures reflect their heterogeneity and polarization in situ. Increased density of CD163+ (CD206−) TAMs with concurrent high CD68 expression is associated with upregulated immune-signaling and improved patient survival by univariate, but not multivariate analysis. CD68-only and CD206+ TAMs are correlated with high PDL1 expression.

[1] Upper Gastrointestinal Translational Research Laboratory, Peter MacCallum Cancer Centre, Melbourne, Victoria 3000, Australia. [2] Sir Peter MacCallum Department of Oncology, The University of Melbourne, Victoria 3010, Australia. [3] Department of Medicine, Royal Melbourne Hospital, The University of Melbourne, Victoria 3010, Australia. [4] Department of Pathology, Peter MacCallum Cancer Centre, Melbourne, Victoria 3000, Australia. [5] The Australian Institute for Musculoskeletal Science (AIMSS), The University of Melbourne and Western Health, St. Albans, Victoria 3021, Australia. Correspondence and requests for materials should be addressed to A.B. (email: alex.boussioutas@petermac.org)

Gastric cancer (GC) is one of the leading causes of cancer-related deaths worldwide[1]. This may, in part, be attributed to different GC subtypes, heterogeneous tumor micro-environments (TME) and/or limited treatment options[2]. GC is commonly described using histological criteria into intestinal, diffuse and mixed subtypes[3]. More recently, an integrated genomics-based approach reported by TCGA[4] revealed four GC molecular subtypes: Epstein-Barr virus-positive (EBV), micro-satellite instability (MSI), chromosomal instability (CIN), and genomically stable (GS), with each subtype demonstrating different TMEs and clinical outcomes[2]. The EBV and MSI GC subtypes are associated with higher degrees of immune signaling and therefore are candidates for immune-checkpoint therapies (ICT), including those targeting programmed death ligand 1 (PDL1)[5]. PDL1 expression has been used to stratify GC patients for ICT, however, is an imprecise biomarker of response[6] suggesting a requirement for more stringent selection criteria.

Upregulated PDL1, detected in approximately 40% of GC cases, has been positively associated with tumor-associated macrophage (TAM) infiltration[7]. TAMs are one of the most abundant immune components in cancers[8] and are characterized by their plasticity and multiplicity of function, contribution to tumor metastasis, immune suppression, and resistance to therapy[9]. However, the role of TAMs in GC is conflicting. TAMs were shown to be related to a stromal-associated gene signature and poor patient outcome[10], but have also been correlated with a high degree of tumor cell apoptosis and good prognosis[11]. This disparity may be associated with their heterogeneity within individual tumors[12,13].

TAMs are a diverse population of cells[14]. The bipolar M1/M2 paradigm which describes the polarization of macrophages in cell culture with lipopolysaccharide (LPS) and interferon-γ (IFN-γ) (M1) or interleukin-4 (IL-4) and IL-13 (M2) has been widely used to classify TAMs[15]. However, accumulating evidence suggests that this is an oversimplification and their complexity would be better described as a dynamic spectrum of phenotypes[16]. Macrophages were shown to exhibit distinct transcriptomes in response to different in vitro stimuli[17] and high-dimensional TAM heterogeneity has been described in different tumor models[18,19]. Advances in single-cell sequencing has significantly improved our understanding of the myeloid compartments[20]. However, due to the limitations of these methods which destroy the tumor architecture, the spatial relationship between different TAM populations in situ remains a significant problem[21].

In the current study, we sought to comprehensively investigate the TAM spatial heterogeneity in GC, and additionally, to determine their relevance to PDL1 and patient survival. A multiplex immunohistochemistry (m-IHC) panel consisting of surface and intracellular markers was designed: CD68, an endosomal/lysosomal glycoprotein that is highly expressed by the mononuclear phagocytes, was used as a pan-macrophage marker[22–24]; scavenger (CD163) and mannose (CD206) receptors are highly expressed by the M2-like macrophages[8,25–27], while interferon regulatory factor 8 (IRF8) is upregulated in M1-like macrophages[28] and was associated with their functions[29–31]; the immune-checkpoint marker PDL1 was incorporated; and AE1AE3 (pan-cytokeratin) was used for identifying tumor cells[32].

Using this panel, we describe seven predominant TAM populations characterized by specific combinations of markers on individual cells. Location, PDL1 expression, and environmental transcriptome signature associated with each population are determined. Our results reveal the spatial distribution of TAM heterogeneity in situ in GC and highlight how TAM characterization with m-IHC may provide further information on macrophage polarization in different tumor anatomic regions. This may assist in the identification of possible therapeutic targets.

## Results

**Experimental definitions and conditions**. To investigate the macrophage landscape within GC, 56 full-face formalin-fixed, paraffin-embedded (FFPE) patient samples were selected (Supplementary Table 1; see Methods section). An H&E-stained tissue section was reviewed by an anatomical pathologist (C.M.) to identify tumor core (C, major body of the tumor mass), edge (E), margin (M), and non-tumor normal (N), which we refer to as regions of interest (ROI). These represent tumor position relative to surrounding tissue (Fig. 1a; see Methods). The serially sectioned tissue was stained with the m-IHC panel (Fig. 1b, c). A total of 1800 high power fields (C: 52%, E: 13%, M: 21%, N: 14%) were imaged across all patient ROI. A supervised image analysis system (inForm[33]) was used to segment each image into tumor-nest[34] (AE1AE3+) and stromal (AE1AE3−) areas (Fig. 1c). In addition, cell phenotyping data were obtained based on the pattern of marker expression (Fig. 1d–f).

**Characterization of macrophage populations**. Macrophage marker expression was analyzed at a single-cell level and seven major populations were characterized and validated (Fig. 1d–f, Supplementary Figs. 1, 2). Criteria for identifying macrophages were CD68 positive and AE1AE3 negative staining. Macrophages were further subdivided based on the positivity and relative intensity of other markers[8,27,35].

Two M1-like TAM populations were identified based on the absence of CD163 and CD206. The CD68+IRF8+ TAMs had high nuclear IRF8 staining, whereas the CD68+ macrophages were only positive for CD68 (negative for other markers). Five M2-like TAM populations were identified by the presence of CD163 and/or CD206. The CD68+CD163+ and CD68++CD163+ TAMs were distinct from each other with the latter population having significantly higher CD68 expression (Supplementary Fig. 1c–e). TAM populations expressing negligible CD163 were characterized using differential expression of CD206, referred to as CD68+CD206++ and CD68+CD206+ (Supplementary Fig. 1f, g). The final population, CD68+CD163+CD206+, expressed both M2-like markers.

The individual TAMs for all patients were plotted based on the intensity of CD68, CD163, and CD206 (Fig. 1g) providing evidence of a spectrum of macrophage populations. The average intensity of each marker was then determined on a per patient basis (Fig. 1h) confirming that these populations are represented within each individual patient sample and ROI (Supplementary Fig. 2).

**Distinct distribution of TAM populations across ROI**. To examine the TAM distribution within the microenvironment, we analyzed the spatial density of the populations, characterized above, within different ROI. A significant increase of the overall macrophage density was observed within the tumor regions (core, edge, and margin) compared with the adjacent normal tissue (Fig. 2a). The TAMs exhibited a more M2-like phenotype at the margin whilst a significant increase in proportion of M1-like TAMs was seen in the core (Fig. 2b). The distribution of each TAM population was then explored (Fig. 2c). The CD68+, CD68+CD206++ and CD68+CD163+CD206+ macrophages were abundant in all ROI but given their dominant presence in the adjacent normal tissue (white circle; Fig. 2c and Supplementary Fig. 3a), they could be associated with the normal-tissue macrophages more than tumor-associated macrophages. CD68+CD163+CD206+ macrophages accumulated at the margin and decreased toward the core. In contrast, the CD68+IRF8+ macrophages increased significantly from the margin into the core which contributed to the shift of the M1-like to M2-like ratio (Fig. 2b and Supplementary Fig. 3b).

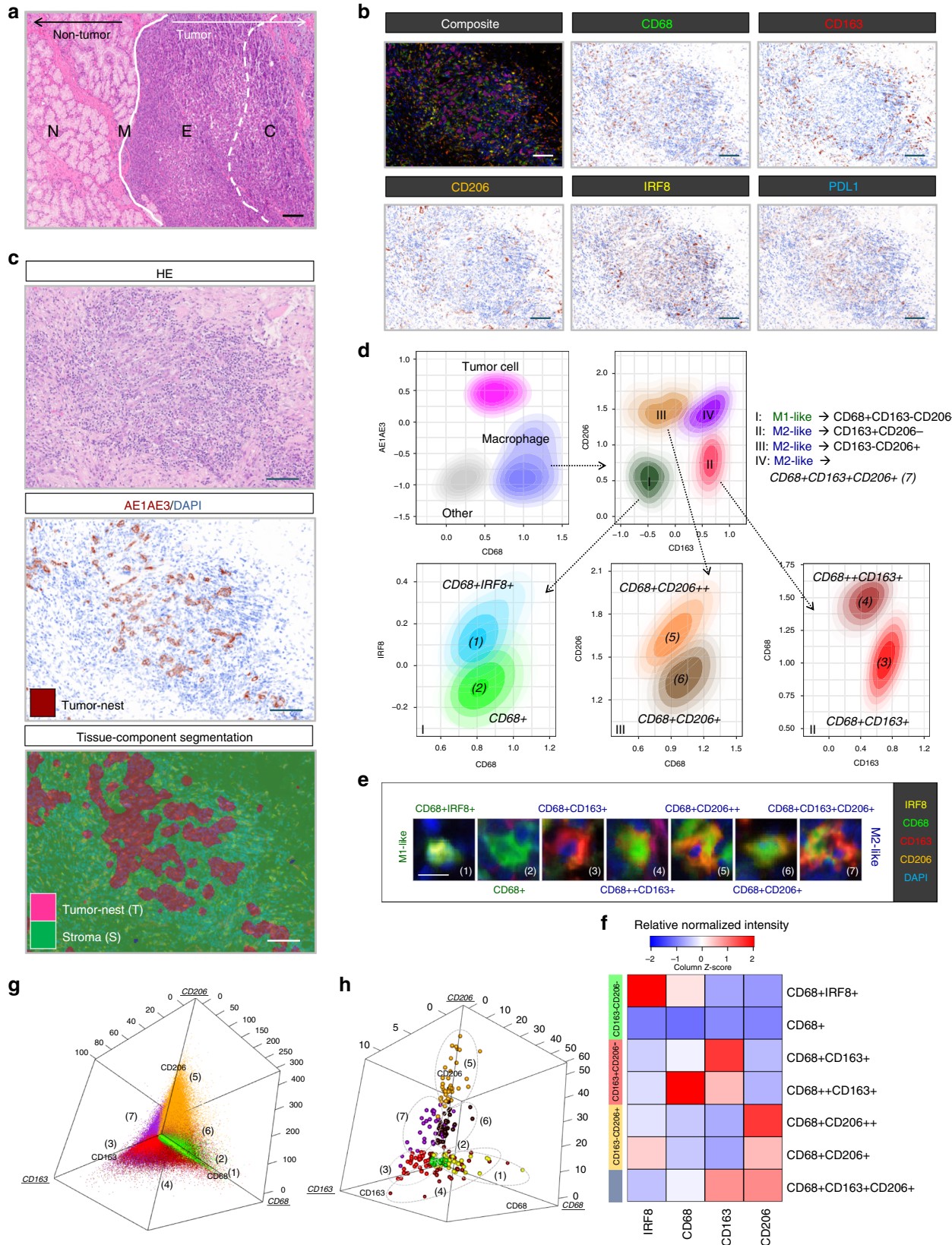

Higher densities of CD68+CD163+, CD68++CD163+, and CD68+CD206+ macrophages were found within the tumor regions when compared with normal tissues suggesting that these populations were polarized according to their location in the tumor microenvironment (Fig. 2c).

The localization of TAMs with respect to the tumor-nest and stromal areas (defined in Fig. 1c) was further examined. The CD68+CD163+CD206+ macrophages were located primarily in the stroma across matched patient samples (Fig. 2d, e). In contrast, CD68+IRF8+ macrophages, were more abundant in the

**Fig. 1** Identification and characterization of macrophage populations. **a** Regions of interest (ROIs): adjacent normal tissue (N), margin (M), edge (E), and core (C). Scale bar: 100 μm. **b** Representative composite and single-stained IHC images of the multiplex IHC panel. Scale bar: 100 μm. **c** H&E, single-stained AE1AE3, and tissue-component segmentation of the same region. Scale bar: 100 μm. **d** Multiplex IHC panel design: gating strategy for each TAM population (numbered). **e** Seven major TAM populations. Positivity (+) of corresponding markers and relative intensity between populations is indicated. Scale bar: 10 μm. **f** Marker signatures used for TAM population characterization in patient samples ($n = 35$). Relative normalized intensity: relative original intensity of each marker divided by exposure time. **g**, **h** 3D plots showing the intensities of TAM populations from (**g**) single cells ($n = {\sim}8.5 \times 10^6$ from 56 patients) and (**h**) averaged per patient ($n = 35$). Unit of axis: Normalized intensity. Key: Orange: CD68+CD206++, Brown: CD68+CD206+, Green: CD68+, Yellow: CD68+IRF8+, Dark red: CD68++CD163+, Red: CD68+CD163+, and Purple: CD68+CD163+CD206+. TAM populations are as numbered in (**d**) and (**e**)

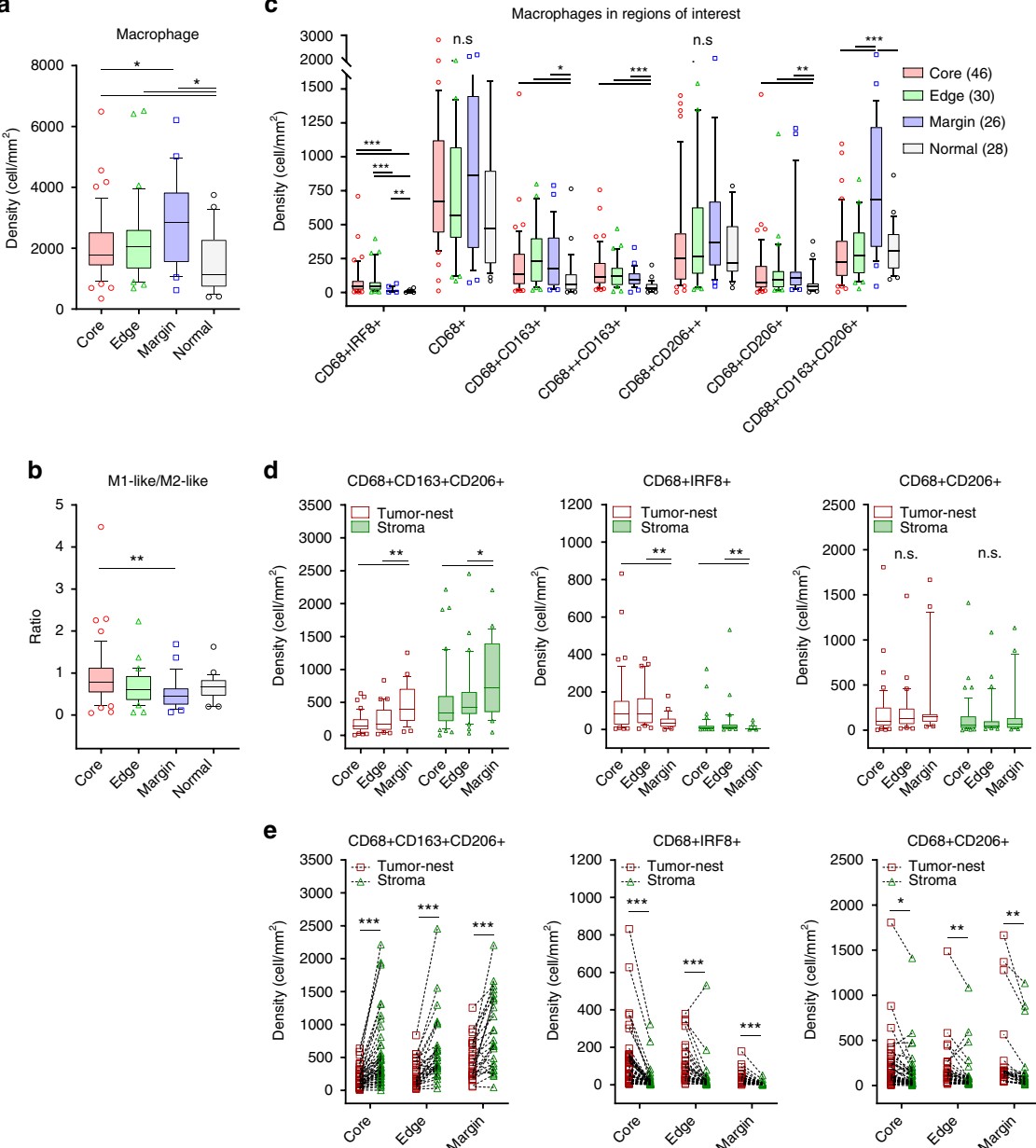

**Fig. 2** Distinct distribution of TAM population densities across regions of interest. **a**–**c** Spatial distribution of TAM populations: **a** Overall TAM density, **b** M1-like to M2-like ratio, and **c** Density of each TAM population. Core (red circle): $n = 46$, edge (green triangle): $n = 30$, margin (blue square): $n = 26$, normal (white circle): $n = 28$. **d**, **e** Density of selected TAM populations between the Tumor-nest (dark red square) and Stroma (dark green triangle) areas (**d**) among the ROIs and (**e**) in matched (dash line) patient samples. Box and whiskers represent mean ± 10–90 percentile. Each point represents one patient. *$p < 0.05$, **$p < 0.01$, ***$p < 0.001$ and not significant (n.s.). Mann–Whitney $U$ test

core compared with the margin (Fig. 2c, d), and were located within the tumor-nest than in the stroma (Fig. 2e). The CD68+CD206+ population was also found to be more specific to the tumor-nest area (Fig. 2e) but did not differ among the ROIs (Fig. 2d). No other quantification of TAM densities were significant between the tumor-nest and stromal areas and were independent of GC subtypes and clinical parameters (Supplementary Fig. 3c–f).

**TAM composition is associated with tumor cells**. The observation that certain macrophage populations were enriched in the tumor region (Fig. 2e), suggested the proximity of TAMs to tumor cells might influence their phenotype. To further study these localization patterns[36], a bioinformatics tool (ISAT; see Methods[37]) which determines the nucleus to nucleus distances between any two cell types was developed (Fig. 3a).

Distances between cells within the tumor core were analyzed and identified the CD68+IRF8+ macrophages as the population located in closest proximity to the tumor cells with a median nucleus-to-nucleus distance of 12.3 µm (Fig. 3b). In contrast, the CD68+CD163+CD206+ macrophages were furthest from the tumor cells (median distance 23.8 µm). The distribution patterns of TAM populations relative to tumor cells at the edge and margin were similar to those in the core (Supplementary Fig. 4a).

To incorporate both cell proximity and quantity, an "effective percentage" parameter was introduced (Fig. 3a). This represents the proportion of macrophages that had a tumor cell within defined distance criteria. A significantly higher percentage of the CD68+CD206+ and the CD68+IRF8+ macrophages had tumor cells within a 10 µm radius (median effective percentages of 31 and 27%, respectively; Fig. 3c and Supplementary Fig. 4b). The CD68+CD163+CD206+ macrophages (12%) were the least associated with tumor cells within this range.

When the distance was extended to within 10–20 µm, the CD68+IRF8+ macrophages (47%) remained the major population associated with tumor cells. The CD68++CD163+ macrophages (40%) were the next most abundant (Fig. 3c and Supplementary Fig. 4b) and the CD68+CD163+CD206+ macrophages remained the abundant population beyond 20 µm from the tumor cell (Fig. 3c, d).

These results indicate that despite being a large population within the tumor, the majority of CD68+CD163+CD206+ macrophages were located at a distance approximately 2–3 average cell-lengths from the tumor cells. The majority of the CD68+IRF8+ macrophages (78%) had tumor cells within 20 µm radius of their nucleus (Fig. 3d). All TAM-tumor pairings peaked within 10–20 µm, a distance which would put cells into direct contact with each other. This suggests that this distance may be the zone for TAM polarization or phenotypic change in situ, and could potentially be used as a threshold for partitioning the tissue for subsequent genomic/proteomic analyses.

Collectively, our data show that the predominant TAM population differs phenotypically between the tumor and adjacent normal tissue (Fig. 2). The same phenotypic differences were observed in macrophages proximal to the tumor cells (Fig. 3b–d).

**Influence of TAMs on patient survival**. To assess the relevance of TAMs to relapse-free survival (RFS) and overall survival (OS), patients were stratified based on the density of TAMs within the tumor core (Supplementary Fig. 5a–c). The core was chosen as it comprises the majority of the tumor mass and also because some tumors, particularly those of diffuse histology, do not have a definitive margin.

In a univariate analysis of outcome, TAMs expressing CD68++CD163+ within the core were associated with improved RFS, but not OS (Fig. 3e and Supplementary Fig. 5d). This observation did not reach statistical significance after multivariate analysis incorporating known clinical prognostic factors (Supplementary Fig. 5e).

The effective density (0–10 µm), which reflected the absolute number of macrophages that had a tumor cell within a 10 µm radius (defined as direct contact), was used as an additional measurement. The results showed that patients with a higher effective density of CD68++CD163+ macrophages had significantly longer RFS and OS (Fig. 3f and Supplementary Fig. 4d) when compared with patients with lower TAM density. No other TAM population was found to be associated with patient survival (Supplementary Figs. 4c–f, 5).

There were no survival differences found if only proximity or cell proportion (median distance, effective percentage; Supplementary Fig. 4c, e) but not cell number (overall density, effective density) was used as patient classifying factors. These results indicate that the influence of CD68++CD163+ macrophages on patient survival could be relevant to both the number of macrophages and their proximity to tumor cells. In summary, our data indicate that to determine the influence of TAMs on tumor cells and patient survival, location in addition to population density, should be taken into consideration.

**Different TAMs co-localize in a same tumor microenvironment**. The co-existence of different TAM populations in the tumor microenvironment has been recognized previously[8,34], but their interaction is not well studied in GC. To understand the possible interaction between these macrophages, first the co-localization of TAMs within the tumor core was investigated. The CD68+IRF8+ macrophages were significantly enriched in an environment with increased CD68++CD163+ and CD68+CD206+ TAMs, but negatively associated with the CD68+CD206++ and CD68+CD163+CD206+ TAMs. The CD68++CD163+ population was further co-localized with the CD68+ and CD68+CD163+ TAM, which was, in turn, significantly related to the CD68+CD163+CD206+ populations (Fig. 4a and Supplementary Fig. 6a).

**Environmental signatures reflect TAM polarization in situ**. Environmental factors which contribute to the dynamic process of macrophage polarization in vivo[38] might be the cause of TAM co-localization patterns. To further study the influence of these factors, the TAM densities in the core were correlated with whole tumor-derived transcriptomic microarray data (Affymetrix U133+ 2)[10] derived from the same patient ($n = 34$) to generate TAM population-specific environmental gene signatures (Fig. 4b).

The genes that were significantly correlated ($p < 0.05$; Spearman correlation) with each TAM density were identified as the signature for each TAM population. The most relevant pathways associated with each signature were further determined using Reactome[39] (Fig. 4b). These signatures comprised of the genes that were population unique and also some shared elements between populations (Supplementary Fig. 6b). Significant pathways (FDR < 0.05; Benjamini-Hochberg[40]) are listed in Supplementary Fig. 6c.

Microenvironments with abundant CD68+IRF8+ TAMs were characterized by an increase in IL-1 signaling and inflammatory/apoptotic cell death pathways (Fig. 4b). The CD68++CD163+ population was characterized specifically by increased IL-10, extra-cellular matrix (ECM) organization, other interleukin signaling pathways and the down-regulation of DNA repair pathways. The CD68+CD206+ population was defined by enrichment in multiple interleukin signaling pathways, especially IL-6. The CD68+CD163+CD206+ population showed

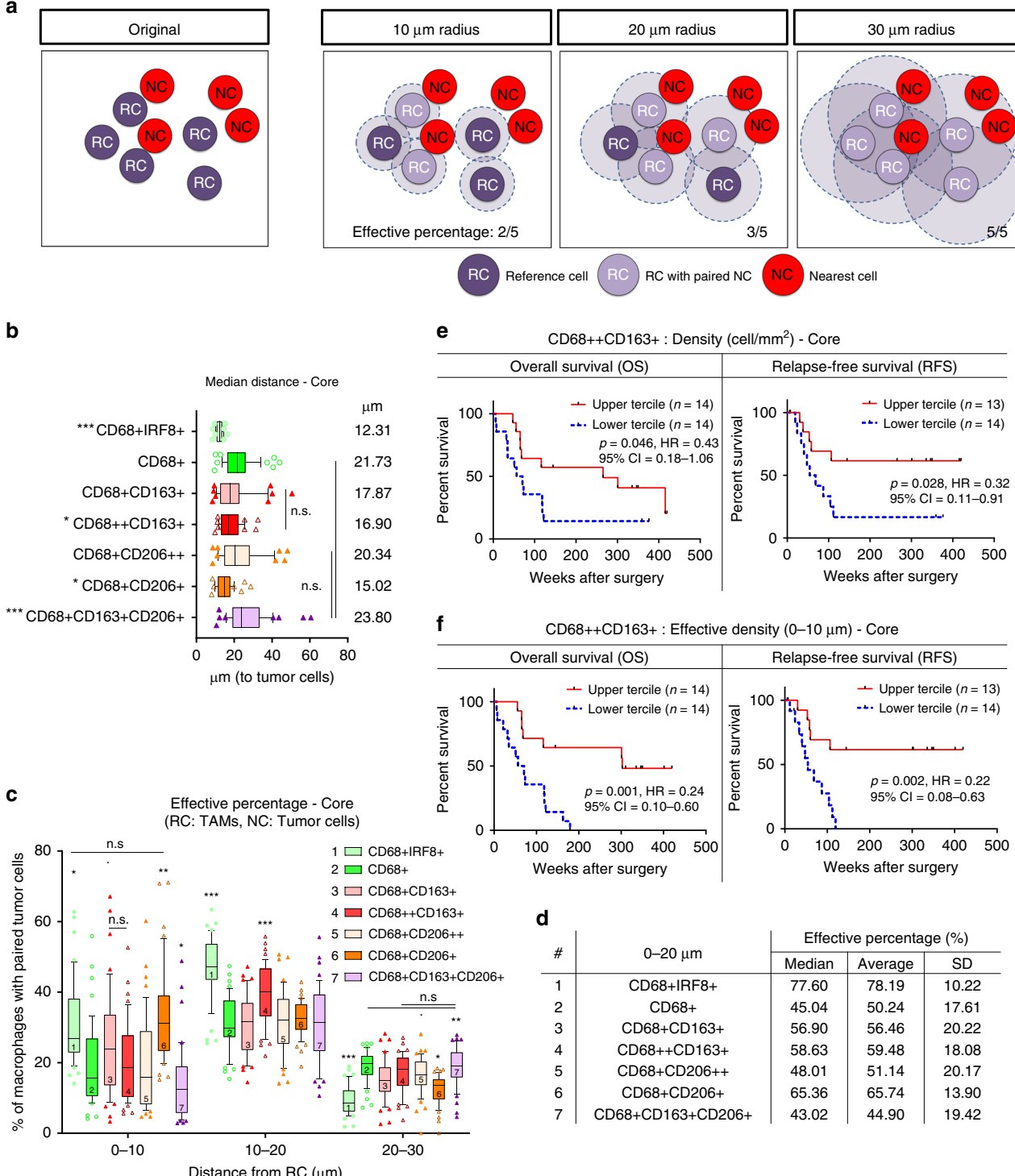

**Fig. 3** TAM composition is associated with their proximity to tumor cells. **a** Schematic illustration of the distance analysis involving reference cell (RC) and nearest cell (NC). Each RC was paired with a neighboring NC in 10 μm increments from its nucleus. Effective percentage represents the proportion of RC that had a paired NC in a given distance. **b** Median distance of TAM populations (RC) to tumor cell (NC). Box and whiskers represent mean ± 10–90 percentile. Each dot represents one patient. Core: $n = 46$. *$p < 0.05$, **$p < 0.01$, ***$p < 0.001$ and not significant (n.s.). Mann–Whitney $U$ test comparing between TAM populations. Circle: M1-like macrophages. Triangle: M2-like macrophages. **c**, **d** Effective percentage of TAM populations in the core (**c**) within 10 μm increments and (**d**) within 0–20 μm. **e**, **f** Overall survival (OS) and relapse-free survival (RFS) classified by the (**e**) overall CD68++CD163+ TAM density and the (**f**) effective density (0–10 μm) in the tumor core. Effective density (0–10 μm): the number of TAM that had a tumor cell within a 10 μm radius. Upper tercile (density > 2/3 of the patients in the cohort; red line), Lower tercile (density ≤ 1/3 of patients in the cohort; blue dash line). Log-rank (Mantel-Cox) test

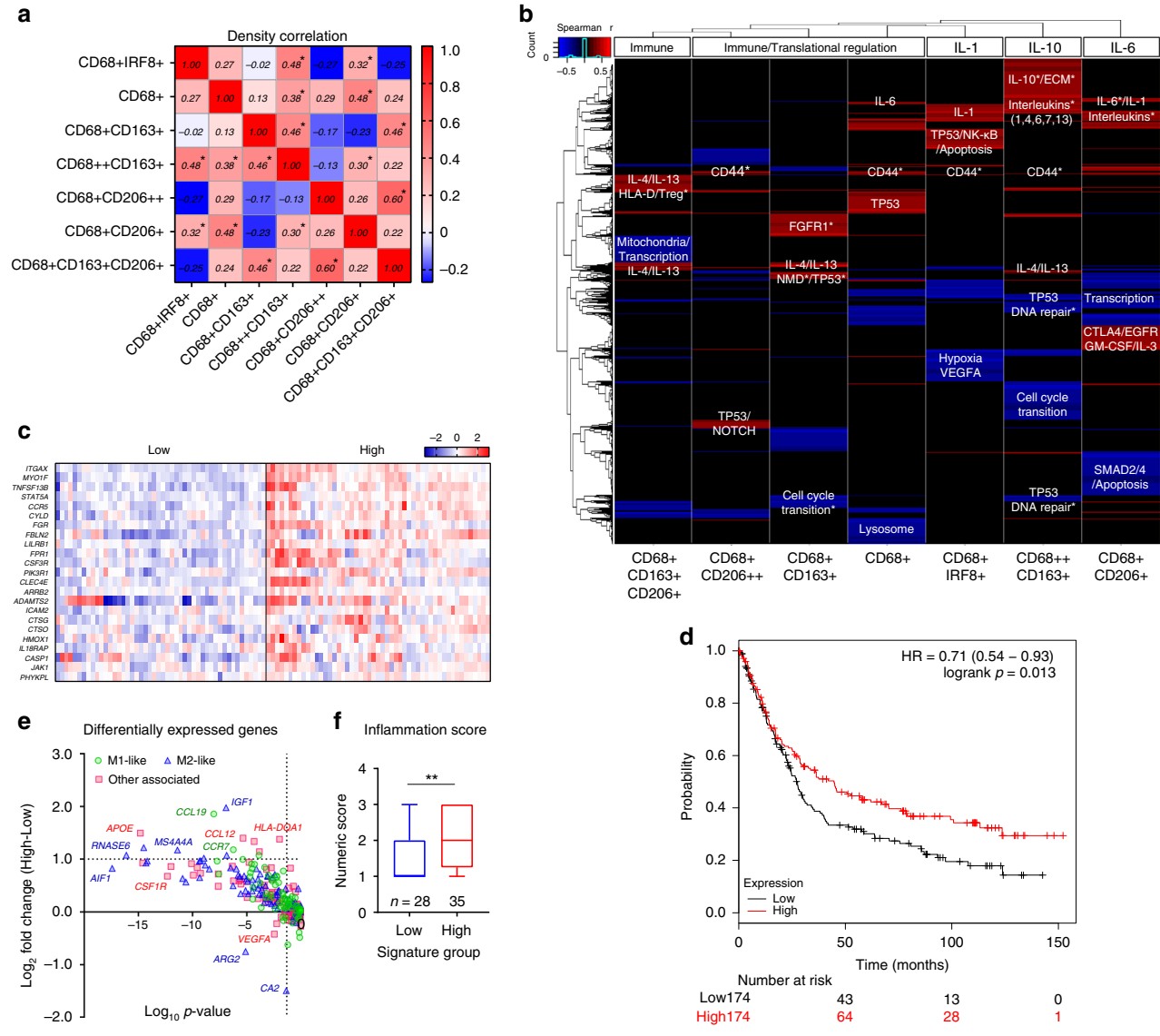

**Fig. 4** High CD68++CD163+ TAM density reflects a more inflamed microenvironment and enhanced immune-cell infiltration. **a** Spearman correlation of TAM population densities in the tumor core. Red/white/blue: positive/no/negative correlation. *$p < 0.05$. **b** Unsupervised clustering of TAM-associated environmental signatures and associated pathways. List of genes that were significantly correlated with each TAM population density of a matched patient sample ($n = 34$) were identified with the Spearman correlation. Red/black/blue: positive/no/negative correlation. *FDR < 0.05 (Benjamini-Hochberg[40]). **c** Patients in the MAUGIC cohort ($n = 99$) grouped using a refined CD68++CD163+ TAM signature (gene expression most significantly correlated with cell density, $p < 0.001$; Spearman correlation). Key: relative gene expression. **d** Validation of patient overall survival classified by the refined CD68++CD163+ TAM signature using KMplot[41]. Above (red) and below (black) the median expression of the gene signature. Log-rank (Mantel-Cox) test. **e** Differentially expressed macrophage-associated genes, and **f** inflammation scores between the high–low patient groups in the MAUGIC cohort. Box and whiskers represent mean ± 10–90 percentile. **$p < 0.01$. Mann–Whitney U test

upregulation in IL-4/IL-13 signaling pathways and genes relevant to the development of regulatory T cells. These results, together with the pattern of density distribution between ROIs (Fig. 2c) and the co-localization pattern (Fig. 4a), may reflect the dynamic process of macrophage polarization in situ.

**High density of TAMs reflects an inflamed microenvironment.** To explore possible reasons underlying the survival advantage in patients with an enriched CD68++CD163+ population, a refined gene signature using the genes that were most significantly correlated ($p < 0.001$; Spearman correlation) with its density within the core was developed (Fig. 4c). This signature was validated using our extended patient cohort ($n = 99$; Fig. 4c,

Supplementary Fig. 6d, e). A publically available online GC survival database ($n = 348$)[41] was then tested, patients with above median level of expression of this signature had an improved overall survival (Fig. 4d).

To investigate the immunological features of patients grouped with the refined CD68++CD163+ TAM gene signature (Fig. 4c), differentially expressed macrophage and other immune-cell related genes were compared. Upregulation of both M1-like and M2-like macrophage and also T and NK cell related genes were observed (Fig. 4e and Supplementary Fig. 6f). This also correlated with increased inflammation scores[42] as determined by a pathologist (C.M.; Fig. 4f) and with high CD3+CD8+ T cell number in the core (Supplementary Fig. 6g). These data suggest

that high CD68++CD163+ TAM density reflects an enhanced host immune response and could be one reason for the improved patient survival seen on univariate analysis.

**High PDL1 expression in CD68-only and CD206+ macrophages**. Inhibition of PDL1 by ICT has been assessed as a potential therapeutic for GC[5]. The data presented in this study show that PDL1 is expressed by all TAMs (Fig. 5a), however, the CD68+CD206+ macrophages had significantly higher mean PDL1 expression per patient compared with the other TAM populations (Fig. 5b). Given the pattern of PDL1 in the tumor microenvironment was related to cell location, cell number and intensity on each cell (Fig. 5a); PDL1 expression on the individual cells across all ROIs was further analyzed. It was observed that all cell types, including macrophages and tumor cells, had a proportion of cells with high PDL1 expression (Supplementary Fig. 7a). Using the mean PDL1 expression across individual cells in the cohort (visible PDL1 staining; Supplementary Fig. 7b) as a threshold of positivity, thirty two percent of all cells (irrespective of cell types) were defined as PDL1+ (Fig. 5c and Supplementary Fig. 7b, c; see Methods).

The number of PDL1+ macrophages (regardless of population) was similar to the number of PDL1+ tumor cells. More TAMs were within the top 10% of the PDL1+ cells in our cohort (Fig. 5d). A more detailed interrogation of the data identified three abundant PDL1+ TAM populations, namely CD68+, CD68+CD206++ and CD68+CD163+CD206+ (Fig. 5d and Supplementary Fig. 7c). To control for differences in the cellular infiltrate (Fig. 2c) that may influence the absolute number, the percentage of PDL1+ cells in each cell type was compared. All three CD206+ TAM populations had at least 36% of cells that were PDL1+ (Fig. 5d and Supplementary Fig. 7c). However, although the percentages of PDL1+ cells within the CD206+ macrophages were similar, their intensity expression patterns were different (Fig. 5e). CD206+ (CD163−) macrophages exhibited most intense expression, whereas the CD68+CD163+CD206+ macrophages were equally distributed among different intensities.

These results showed that the CD68-only (high in number) and the CD206+ TAMs (high in number and percentage of cell type) were the main PDL1-expressing populations in our cohort. The high mean PDL1 expression on CD68+CD206+ observed per patient was due to having a larger proportion of PDL1+ cells (Fig. 5d).

**PDL1 expression on TAMs is associated with cancer subtype**. To associate PDL1+ TAMs with different GC subtypes, we identified the TAM population with the highest PDL1 intensities from each patient. PDL1-high patients ($n = 44$) were categorized by having any TAMs represented in the top 1% of the total cell population (Fig. 5c and Supplementary Fig. 7b). Among the PDL1-high patients, the CD68+ macrophages were predominant in the GS and diffuse subtypes, whereas the CD68+CD206++ macrophages were enriched in the MSI and intestinal subtype (Fig. 5f). In addition, TAMs in the GS and diffuse cancer subtypes had significantly lower median PDL1 expression (Fig. 5g). Whilst we found a significant association between PDL1+ TAM in GC subtypes, we did not find substantial survival differences using TAM-PDL1 expression (Supplementary Fig. 7d). Indeed, the PDL1 expression was more predictive of cancer subtype than TAM composition (Supplementary Fig. 7e).

**High PDL1+ TAMs are located in the tumor-nest**. From the results above, TAM populations with the highest PDL1 expression were identified. Using the tumor core as the region of

interest, the location of these PDL1+ TAMs was further investigated by comparing the expression changes of PDL1 on TAMs at the interface between the tumor-nest (T) and stroma (S) areas (TS; Fig. 6a, b).

The mean PDL1 expression defined the threshold for PDL1 positivity as described in Fig. 5c. Using this threshold, the level of PDL1 expression by tumor cells (AE1AE3+) was at the threshold and the other non-macrophage cells (Other) were negative (Fig. 6b). Most TAMs expressed PDL1 below this threshold in the stroma but showed a progressive increase of PDL1 with increasing proximity to the tumor cell. The CD163+ and CD68+CD206+ macrophages were found to continuously upregulate PDL1 as they became more embedded in the tumor-nest. The CD68+CD206+ macrophage was the only PDL1+ population detected in the stroma (Fig. 6b and Supplementary Fig. 8a). We found a similar pattern of PDL1 expression in other ROIs and confirmed lower PDL1 expression in the GS and diffuse cancer subtypes (Supplementary Fig. 8).

**Macrophage marker expression differed between tumor areas**. Markers expressed by macrophages often reflect their function[8]. The change in macrophage markers at the TS interface was investigated using the mean expression of CD68, CD163 and CD206 of tumor (AE1AE3+) and non-macrophage (other) cells as baseline thresholds (Fig. 6c–e).

The CD163+ and CD68+ IRF8+ macrophages exhibited a minimal twofold increase in their CD68 expression between the TS areas and this was induced ~25 μm before engaging the TS interface (Fig. 6c). The CD206+ (CD163−) and CD68-only TAMs had relatively little changes. The results suggested that the CD163+ and CD68+IRF8+ macrophages may differ in their phagocytic abilities[22] compared with the CD206+ (CD163−) macrophages and the induction of CD68 did not require direct contact between the TAMs and the tumor cells.

With increasing proximity to tumor cells, an increase in CD163 expression was observed in all CD163+ macrophages (Fig. 6d). Interestingly, unlike CD68 and CD163, the expression of CD206 on each TAM population was not differential across the TS areas and only the CD68+CD206++ macrophages showed slightly increased expression within the tumor-nest (Fig. 6e). However, the CD68+CD206+ TAMs were predominately located within the tumor-nest (Fig. 2e) suggesting that the overall CD206 expression on the CD206+ (CD163−) macrophages (CD68+CD206++ plus CD68+CD206+) decreases as they near the tumor cells.

## Discussion

Characterization of TAMs in GC has been complicated due to their adaptive changes to environmental stimuli[38], the lack of exclusive markers between populations[43] and the differences between human and animal models[44]. Hence, despite intensive investigation on macrophage heterogeneity[9,20], the distribution of different populations in situ remains unclear in humans. Accumulating evidence suggests that using a combination of markers would be a more reliable approach to distinguish between populations or activation states[21]. Techniques to study TAM populations include IHC[27,45,46], CyTOF[18], flow cytometry[35,47], and single-cell sequencing[20,48]. While the multiplex IHC utilized in this study was limited to using a few cell markers, its strength was its ability to spatially resolve TAM heterogeneity.

Here we describe the use of multiplex IHC on human GC tissues to characterize TAM populations in a spatial context and ultimately their association with clinical outcomes based on the proximity of TAMs to tumor cells. We have identified TAM

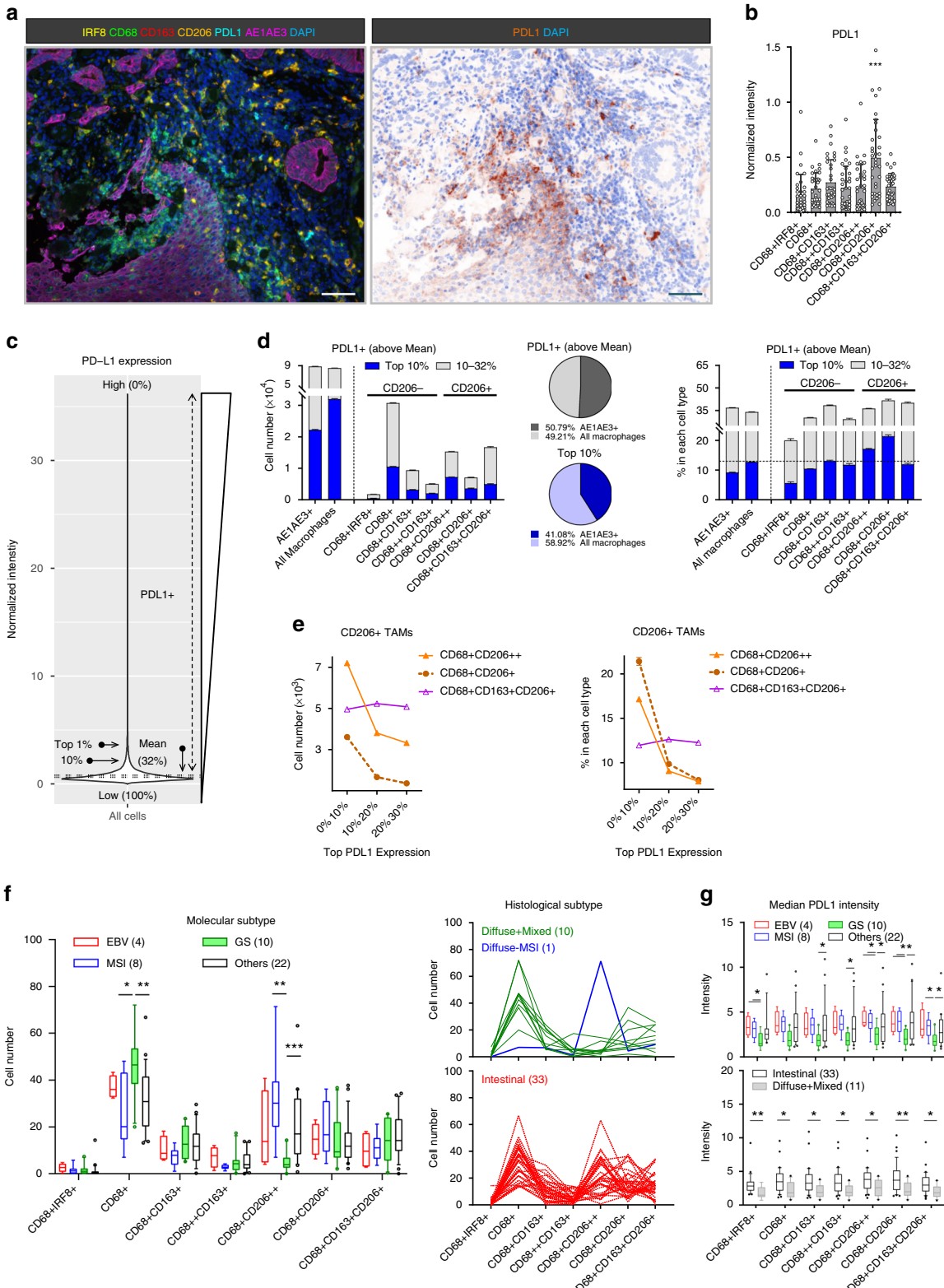

**Fig. 5** PDL1 expression in GC is both TAM and GC subtype associated. **a** Representative images of PDL1 expression in the tumor core. Scale bar: 100 μm. **b** Mean PDL1 intensity per patient ($n = 35$) for each TAM population. Error bars represent mean ± SD. **c** Distribution of PDL1 expression on individual cells in a sampling cohort regardless of cell types. Equal number ($10^4$) of cells per patient ($n = 56$, all ROIs included) were randomly selected to normalize between sample sizes. Cells were pooled as a sampling cohort. 32% of the cohort was defined as PDL1+ with the mean expression of the cohort. **d** Cell number and percentage of different PDL1 intensity expressing cells in each cell type. Error bars represent mean ± SD of five independent randomly selected subsampling cohorts. **e** PDL1 expression patterns of CD206+ TAMs: Cell number and percentage in each population. Error bars represent mean ± SD. $n = 5$. **f** Cell number and **g** median PDL1 expression on TAM populations in patients grouped by different GC subtypes. Box and whiskers represent mean ± 10–90 percentile. Each point/line represents one patient. *$p < 0.05$, **$p < 0.01$, ***$p < 0.001$, and not significant (n.s.) Mann–Whitney $U$ test

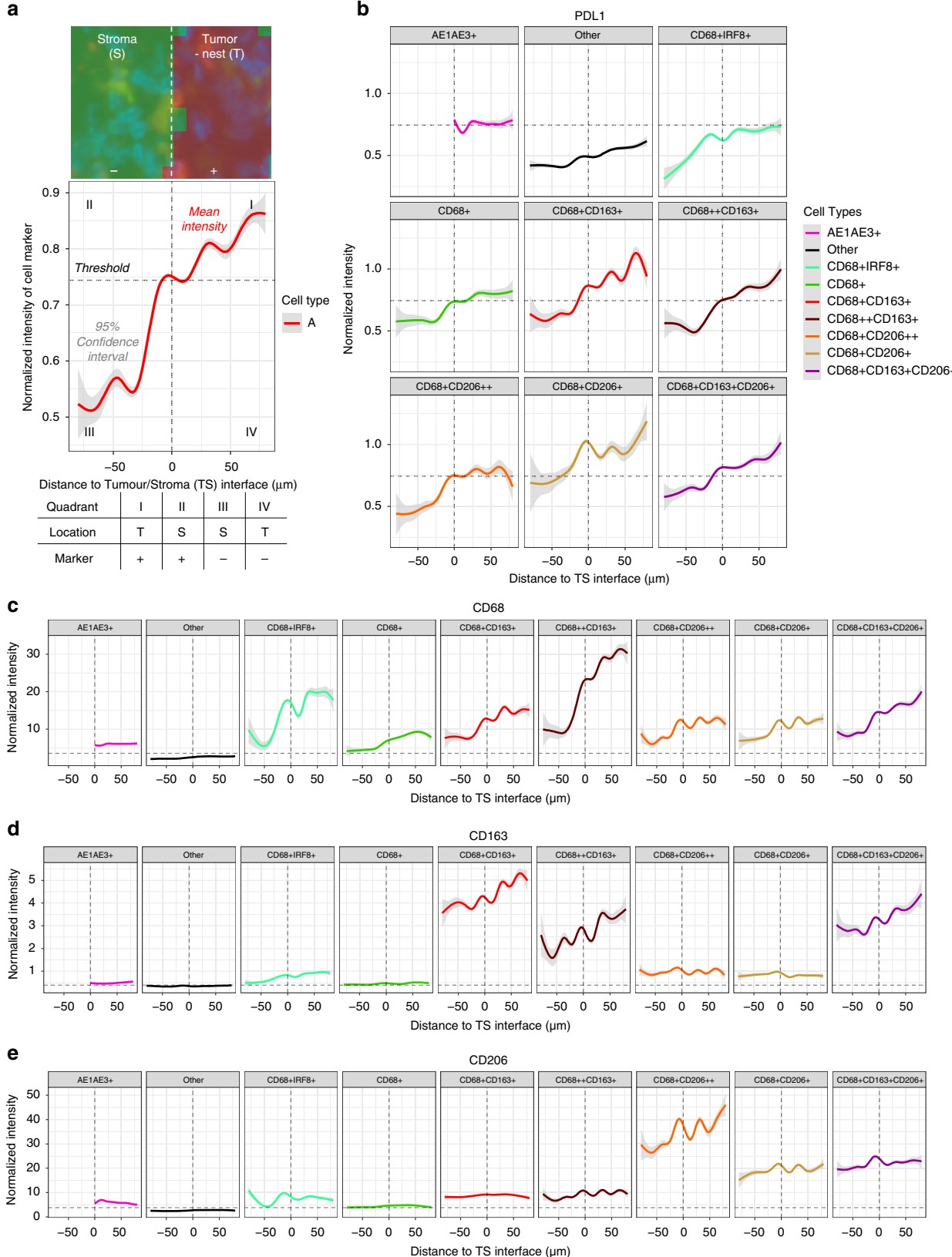

population-specific environmental gene signatures and pathways that provide some information on the influence of the micro-environment on these TAM populations and their putative functions (Fig. 7).

We report seven predominant TAM populations distributed in a nonlinear spectrum based on the staining of four markers. The majority of the CD68+IRF8+ TAMs (78%) were within 20 μm of tumor cells. This proximity may reflect the functional gradient of

**Fig. 6** Macrophage marker expression differed between the tumor-nest and stroma. **a** Tumor-nest (red, T) and stroma (green, S) areas in each image were segmented using inForm software (see also Fig. 1b). Intensity change of markers between the TS regions was determined by randomly assigning equal number of cells (n = 500, minimum number available between samples) from both areas per patient (core). The marker of interest was plotted with the mean (red) and the confidence intervals (gray). The interface between TS regions was defined as point zero, positive and negative values on the x-axis indicate the tumor cell and the stroma region, respectively. Thresholds were applied to define the positivity of markers. Quadrants: I: cell type A located in the tumor-nest and is positive (+) of the marker tested. II: stroma, positive. III: stroma, negative (−). IV: tumor, negative. **b** Change in PDL1 expression with distance from the TS interface for each macrophage population. Threshold: mean PDL1 expression defined in Fig. 5c. **c–e** Change of (**c**) CD68, (**d**) CD163, and (**e**) CD206 expression with distance from the TS interface. Thresholds: mean expressions of CD68, CD163, and CD206 on the tumor and non-macrophage cells (Other)

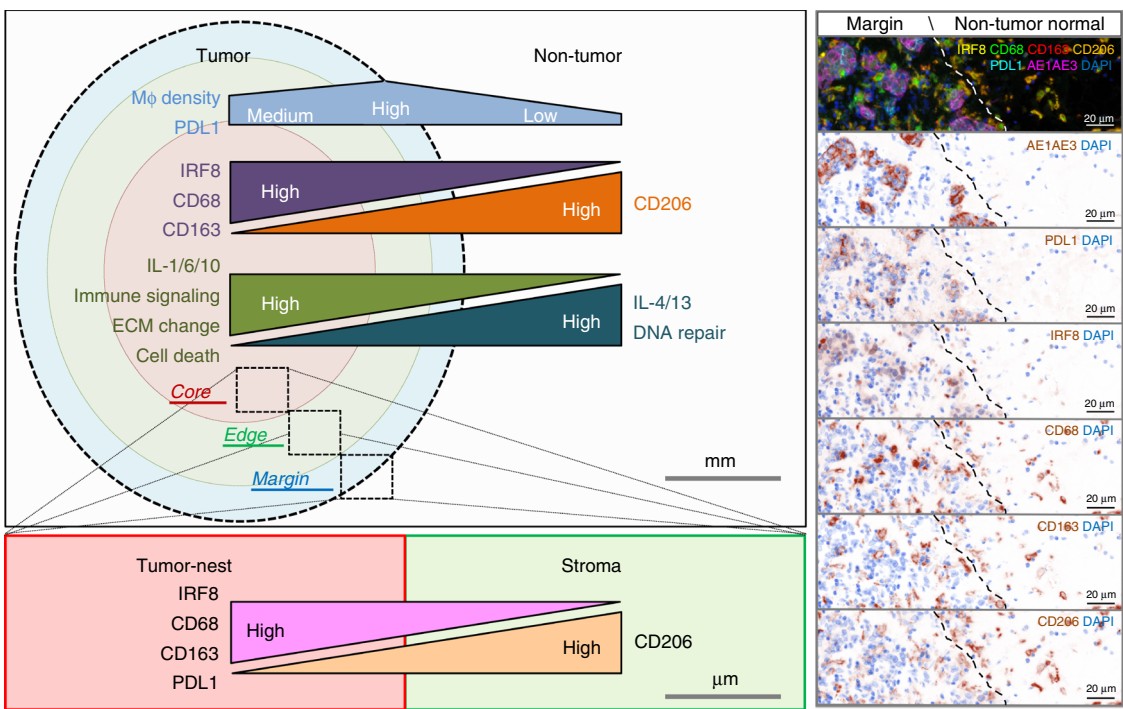

**Fig. 7** Schematic model of spatial macrophage heterogeneity in anatomic regions in gastric cancer. A proposed model suggesting that TAM density and marker expression varies between GC sample regions. Distinct marker expression profiles on TAMs were associated with their population disparity between the tumor site and non-tumor tissue and between the tumor-nest and stroma. TAM populations were associated with signature pathways which may help inform of their function and polarization status. Similar trend of marker changes on TAMs were observed between the tumor-nest and stromal compartments within different tumor regions (core, edge, or margin). Mϕ: macrophage. ECM: extra-cellular matrix

cytokines such as interferons[28] which are thought to trigger M1-like macrophage polarization but also suggests that these TAMs may function through direct contact with a tumor cell. Microenvironments enriched with this TAM population were characterized by the increase of IL-1 related genes and cell death signaling. However, these M1-like TAMs did not significantly impact on patient survival which may be due to their relatively low number compared with the other M2-like TAM populations.

Surprisingly, the CD68+CD163+CD206+ macrophages, expressing both M2-like markers, were abundant in the tumor regions but were consistently located furthest from the tumor cells. This suggests that they may not be the main population affecting the tumor cells directly. Our data show an inverse relationship between CD163 and CD206 expression on TAMs as they near the tumor cells. The CD68+CD163+CD206+ population may include some cells that are of a transient phenotype between the CD163+ (CD206−) and CD206+ (CD163−) populations and also some unpolarized macrophages that co-express CD163 and CD206[43].

The effects of TAMs on GC patient outcome in the existing literature are conflicting. These studies[25,26] invariably use either single staining of CD68 or dual staining in combination with either CD163 or CD206 to identify TAMs using the useful but oversimplified bipolar model[49]. Our data support recent publications advocating the reassessment of using bipolar classification[15,17] and our findings suggest that the presence of CD68+CD163+CD206+ TAMs could possibly dilute out the contributions of smaller populations which may have prognostic value, such as the CD68++CD163+ TAMs.

In a univariate analysis we found that clinical outcome was associated with TAM number and their proximity to tumor cells, highlighting the importance of investigating the distribution of cells. This proximity to tumor cells suggests that cell–cell contact and/or paracrine effects are key to their function.

Interestingly, the high number of CD68++CD163+ TAMs in close proximity to the tumor cell resulted in improved prognosis of patients in our cohort. Although we are unable to describe a cell-specific mechanism to explain this observation, clues may be found from the environmental signatures associated with this TAM population and the nature of the markers expressed. Functionally, CD68 has been shown to be the receptor for apoptotic cells and may be involved in antigen processing[22]. CD163 is a high affinity haptoglobin-hemogobin and HMGB1 scavenger receptor[50] and has been found to be

upregulated on macrophages polarized by IL-10, but not IL-4[51], which is consistent with our findings of higher IL-10-associated signaling pathways in cases where CD68++CD163+ macrophages were abundant. CD163+ macrophages have been shown to localize at perivascular areas[52], clear apoptotic debris, and engage in endocytosis[53]. This evidence suggests that the CD68++CD163+ TAMs may function in the clearance of dead cells, tissue remodeling and anti-inflammatory processes. High numbers of this population near the tumor cell may reflect an enhanced immunological response in these particular tumors.

While high CD68++CD163+ TAM density proximal to the tumor correlated with improved survival in univariate analysis, this was not confirmed in a multivariate model incorporating known prognostic clinical parameters (e.g., age, tumor stage). Larger cohorts are required to confirm our observation.

The interactions between TAM populations have been relatively difficult to determine compared with other more established cell types (e.g., T cells[36]). The diversity of both pro- and anti-inflammatory cytokines in a shared location can contribute to their heterogeneity[54] and the co-existence of different populations might not just reflect the interaction between terminally differentiated subtypes but also reflect the plasticity inherent in macrophages and the change from one phenotype to another[8,55]. For example, IL-1β and IL-6 are produced by pro-inflammatory macrophages[56]. Low level of autocrine IL-10 from these macrophages can serve as a self-protective mechanism[57,58], which can potentially polarize adjacent macrophages into M2-like phenotypes. In our data, we found the CD68+IRF8+, CD68+CD206+, and CD68++CD163+ macrophages to be co-localized in the same tumor and were correlated with the increase of IL-1-, IL-6-, and IL-10-associated pathways, respectively. These results suggest that the co-existence of different TAM populations could resemble their dynamic polarization process in tissue[59], and when more detailed TAM subgroupings were applied, this co-localization data could potentially be used for the deconvolution of whole tumor-derived gene expression data. In addition, it would be difficult to determine the balance of cytokine and chemokines in the TME even by staining every possible target, but the detection of TAMs may be more pragmatic.

Collectively, our results show that CD163 and CD206 clearly stain different TAM populations and that not all M2-like marker-expressing macrophages might be affecting tumor cells directly[34]. The compositions of TAM populations change not only in millimeter scale between the tumor site and the adjacent normal tissue but also within microns of the tumor cells. These phenotypic differences are due to the relative changes in the expression of CD68, CD163, and CD206 at different distances from the tumor cells and could be associated with different environmental stimuli, possibly related to the metabolic activities in different regions[55] and the propagation threshold of cytokines within the tissue[60,61]. Conclusively, our data demonstrate that TAMs in situ are not just randomly distributed but are influenced by their proximity to tumor cells and the tumor microenvironment.

Consistent with previous reports[4], PDL1 was more highly expressed in the EBV and MSI GC subtypes in our cohort. PDL1 expression was not restricted to a specific cell type but our results showed that PDL1+ TAMs accounted for around 50% of all PDL1+ cells and exhibited some of the highest PDL1 expression. The CD68-only and CD206+ macrophages were identified to be the major PDL1+ populations. These findings raise the possibility of combining checkpoint inhibitors with macrophage targeting strategies in future immunotherapy studies for GC. Given our finding of specific high PDL1 expression on the CD206+ TAMs, the combination of PDL1 and CD206 could potentially be utilized

as an alternative biomarker to PDL1 alone, to identify and stratify patients as candidates for ICTs[6,62]. Moreover, we found PDL1 expression by all TAMs, including the M1-like populations, increased when they were located within the tumor-nest. This result suggests that, despite the fact that the number and proportion of PDL1+CD68+IRF8+ TAMs were the least among all TAMs, their close proximity to tumor cells may confer immunosuppressive properties to other immune cells. This data highlight the complexity of the role of TAMs in GC in vivo where a spectrum of TAM populations coexist, and may represent plasticity that does not easily reconcile with the in vitro M1/M2 model.

There are certain limitations to an m-IHC study and the sensitivity and specificity of the antibodies used is a limiting factor. Whilst CD68 is a recognized mononuclear phagocyte marker, low CD68 expression can be found on other cell types[22]. As a result, the CD68-only population is likely to contain a proportion of non-macrophage cells. In addition, PDL1 positivity may differ between different antibody clones[62]. This provided the rationale for not only defining PDL1 expression with an arbitrary threshold but to analyze the intensity of staining of individual cells. Third, due to the limitations associated with the number of markers that can be used per sample, the number of TAM populations identified in our study was restricted compared with studies using other techniques[18,19]. This number can be further refined if additional markers are incorporated, therefore, we do not state that we defined the optimal grouping for TAMs but investigated the spatial distribution of the populations we can distinguish. Last, whilst we were able to characterize the heterogeneity of TAM populations by associated environmental gene signatures, this was in the context of whole tumor tissue, where cells of gene expression origin are unknown. Further work is required to investigate the origin of these genes for more specific therapeutic implications. This will require single-cell isolation from fresh samples and detailed TAM phenotyping.

In conclusion, our results outline the spatial resolution of macrophage heterogeneity in gastric cancer. We describe the different environmental gene signatures which may reflect the interactive process between macrophage populations in situ and identify the CD206+ macrophages to be most relevant to high PDL1 expression. Our data demonstrate that the heterogeneity of macrophages within the tumor is present at both macro- and micro-levels due to the gradient change of different markers. We emphasize the importance of using high resolution characterization to investigate the roles of macrophage populations in a tissue setting, to identify potential therapeutic candidates and to understand the immune landscape of gastric cancer.

## Methods

**Study cohort and selection criteria**. The Molecular Analysis of Upper Gastro-Intestinal Cancer (MAUGIC) cohort consists of 250 cases of gastric and esophageal cancer patients from 1999 to present (This study was restricted from 1999 to 2009[63]). Written informed consent was obtained from all patients prior to sample collection. All procedures were ethically approved by the Individual Review Boards (IRB) of the Peter MacCallum Cancer Centre and at each of the collection centers.

Specimens from 56 GC specimens from different patients representing all GC cancer subtypes and with comprehensive clinical information were selected. All patients had undergone surgical resection and had at least 10 years follow-up. Eleven cases comprised the phenotyping algorithm training cohort (inForm, PerkinElmer, Massachusetts, USA) and 35 (training cohort included) were used for optimization. All 56 patients were analyzed independently. Three patients with immediate surgery-associated deaths and two with no recurrence data were excluded for overall-survival and relapse-free survival analysis, respectively.

**Classifications of cancer subtype and inflammation status**. GC tissues were classified by an anatomical pathologist (C.M.) into intestinal, diffuse, and mixed types[3]. In situ hybridization was used to determine EBV burden (EBV Early RNA, Roche). MSI cases were identified using IHC for MLH1 (Leica, ES05, 1:50), PMS2 (Ventana, EPR3947), MSH2 (Ventana, G219-1129), and MSH6 (BD Biosciences,

44, 1:800). Tumors showing loss of MLH1 and PMS2 expression, with retention for MSH2 and MSH6 were regarded as microsatellite unstable. Patients classified as diffuse but non-EBV and non-MSI were assumed as GS. The remaining patients were termed "others" and not CIN due to lacking supporting evidence. Degree of inflammation of each GC tissue was scored by a pathologist (C.M.) based on the infiltration of immune cells.

**Multiplex IHC staining protocol**. Opal 7-colour kit (PerkinElmer, NEL811001KT) was used for multiplex IHC. Four micrometers of FFPE sections were dewaxed and rehydrated. In the first round antigen was retrieved with a pressure cooker (EDTA pH 8.0) at 125 °C for 3 min. Slides were cooled to room temperature (RT), washed with TBST/0.5% Tween (3 times, 5 min) and incubated with $H_2O_2$ (3%) for 10 min. Slides were washed and blocked with blocking buffer (Dako, Glostrup, Denmark) for 10 min. Primary antibody, CD163 (Cell Marque, MRQ-26, 1:500, dye540), was incubated at RT for 30 min. Slides were washed and an HRP-conjugated secondary antibody was incubated at RT for 10 min. TSA dye (1:50) was applied for 10 min after washes. This was repeated five more times using the following antibodies, CD68 (Leica Biosystems, 514H12, 1:100, dye570), CD206 (Abcam, Ab64693, 1:6000, dye620), IRF8 (Santa Cruz, E-9, 1:3000, dye650), PDL1 (Spring Bioscience, SP142, 1:2000, dye520), and multi-cytokeratin (Leica Biosystems, NCL-L-AE1/AE3, 1:200, dye690). For second and subsequent rounds antigen retrieval was performed in EDTA (pH 8.0) buffer using a microwave (100–150 mW, 15 min). Nuclei were stained with DAPI (PerkinElmer) and mounted with medium (HardSet, Vectashield). Secondary antibodies anti-rabbit (PerkinElmer, NEF812001EA) or anti-mouse (PerkinElmer, NEF822001EA) were used at a 1:1000 dilution.

**Regions of interest (ROIs)**. The interface of tumor and normal tissue was identified by a pathologist (C.M.). Definitions of the ROIs are as follows: Normal adjacent to tumor (N): the area within the specimen but not within the tumor. Margin (M): the area at the interface of the tumor and normal tissue. Edge (E): the area from interface into the tumor (approximately 1–1.5 mm; depth defined by the limited size of the microscopy field). Core (C): the rest of the tumor.

**Multiplex IHC imaging and inForm analysis**. Slides were imaged using a Vectra microscope. Whole slide scans were performed using the ×10 objective lens. ROIs were selected with fixed-size stamps in Phenochart (PerkinElmer), based on the previously acquired whole slide scan images. 1 × 1 (669 × 500 μm; ×20 object lens) stamp was used for the Margin and 2 × 2 (1338 × 1000 μm) for the Core, Edge, and Normal. As many viable regions as possible in each specimen were selected with minimal overlap. Acquired images ($n = 1800$) were analyzed with inForm for tissue-component segmentation of tumor-cell (AE1AE3+) and stroma (AE1AE3−) regions and cell phenotyping. Density of cells in each ROI was calculated by combining the cell counts from all images and normalizing by the total area (cell/$mm^2$).

**Robustness of TAM population phenotyping**. TAM population phenotyping robustness was tested by randomly subsampling ($n = 2000$) equal number of cells ($n = 100$) from each of the populations per patient within each ROI (Core, Edge, and Margin). The combined-marker signature of the subsampled cells was compared with the signature of the bulk population (Pearson correlation). The number of TAM populations was tested using the K-means clustering method[4] (Fig. S2).

**ImagePro image analysis**. Image output from inForm showing the DAPI, AE1AE3, CD68 and CD163 or CD206 were validated with the Line profile function in ImagePro (Media Cybernetics).

**R analysis**. An Inter-cellular Spatial Analysis Tool (ISAT[37]) was developed using the R software (version 3.3.1 for Windows) for the distance analysis. Distance between two cell nuclei was calculated using the $x$ and $y$ coordinates from the inForm raw data. Each cell of the same phenotype was used as a reference cell to calculate its distance to the nearest cell of different phenotypes. The effective percentage of a cell type was calculated by counting the number of cells within the cell type that had the nearest distance which that fulfilled the distance criteria and normalized to the total number of cells in that cell type. The effective density was calculated by using the number of cells within the cell type that that fulfilled the distance criteria and normalized to the area of tissue ($mm^2$). 10 μm (nucleus to nucleus) was defined as an estimated direct contact distance between the cells.

For PDL1 single-cell expression analysis, to determine a uniform threshold for PDL1 positivity across patients and to normalize for different cell number between patient samples for further analyses, five test cohorts were sampled with equal number of cells ($10^4$) per patient. Cells were randomly assigned using the "sample_n" function of the "dplyr" package.

**Microarray**. RNA from fresh-frozen GC tissue ($n = 99$) collected at the time of surgery was isolated with Trizol (Invitrogen) and column chromatography (RNeasy, Qiagen). Microarrays were hybridized using U133+ 2 chips (Affymetrix)

and scanned with the Genechip Scanner (Affymetrix). Data were previously submitted to Gene Expression Omnibus (GEO; Series GSE51105[10]).

**Environmental gene signatures**. Patients with available core region for multiplex IHC and whole tumor microarray data available were selected ($n = 34$). Environmental gene signatures of each TAM subtype was generated by correlating the cell density results with the gene expression data ($p < 0.05$; Spearman correlation, assuming nonlinear data distribution). Most relevant pathways associated with each signature were identified with the Reactome database. A refined CD68++CD163+ TAM signature was generated using the genes within the significantly associated pathways (FDR < 0.05; Benjamini-Hochberg) and most highly correlated with cell density ($p < 0.001$; Spearman correlation).

**KMplot survival analysis**. Patient survival was interrogated with the refined CD68++CD163+ TAM gene signature using the online database KMplot[41] survival of the MAUGIC dataset (GSE51105) and a combined cohort (GSE14210, GSE15459, GSE22377, GSE29272, and GSE51105) were interrogated.

**Statistics**. GraphPad Prism 7.0 software (GraphPad Software, San Diego, USA). Mann–Whitney $U$ test (two-tailed), Spearman and Pearson correlation, Chi-square analysis and Kaplan–Meier analysis (Log-rank, Mantel-Cox test) were used as appropriate. $P$ values lower than 0.05 were considered as significant.

**Reporting summary**. Further information on research design is available in the Nature Research Reporting Summary linked to this article.

## Data availability
Source data for figures [Fig. 1a–h; Fig. 2; Fig. 3b–f; Fig. 4a, c, e, f; Fig. 5a, c–f, Fig. 6] are provided with the paper. Microarray data are available online (GSE51105). Other data that support the findings of this study are available from the corresponding author [A.B.] upon reasonable request.

## Code availability
The Inter-cellular Spatial Analysis Tool (ISAT) package for R used in the study is available online[37].

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

## Acknowledgements

We thank Dr. Paul Beavis and A/Prof. Paul Neeson for helpful discussions for the study and for reviewing the paper. We also thank Prof. David Bowtell for reviewing the paper, Dr. Sharon Pattison for digitalizing clinical information and conducting patient follow-up, the core facilities at the Peter MacCallum Cancer Centre for supporting data collection and analyses, and all patients and staffs involved in the project. Y.H., M.W. and Y.S. are supported by the Australian Commonwealth Government Research Training Program, the University of Melbourne. Y.H. is supported by the Government Scholarship to Study Abroad (GSSA) issued by Taiwan Ministry of Education. M.W. and Y.S. are supported by the Cancer Therapeutics CRC.

## Author contributions

Experimental work: Y.H., N.D., and C.M.; R Script: Y.H., M.W., and Y.S.; Data interpretation: Y.H., A.A., J.A.H., R.A.B., and A.B.; Writing and editing of the paper: Y.H., N.D., J.A.H., R.A.B., and A.B.; Study concept: Y.H., A.A., J.A.H., R.A.B., and A.B.

## Additional information

**Competing interests:** The authors declare no competing interests.

