## [Peer Review File · Nature Communications]

EDITORIAL NOTE: Parts of this file have been redacted.

Reviewers' comments:

Reviewer #1 (Multiplex IHC)(Remarks to the Author):

Reviewer Comments (NCOMMS-18-38415):

Summary: In their manuscript submission, titled "Macrophage Spatial Heterogeneity in Gastric Cancer Defined by Multiplex Immunohistochemistry" the authors highlight the challenge of TAM heterogeneity in GC and sought to determine the relationship between TAM cellular heterogeneity and spatial organization in terms of PD-L1 expression. The authors propose to resolve TAM heterogeneity by spatial mapping of TAM according to four markers – CD68, CD206, CD163 and IRF8.

Originality and interest: Overall, this submission demonstrates significance and originality. This manuscript contributes towards efforts focused on resolving TAM heterogeneity. I think it's important to clarify the language in the abstract and introduction to highlight that resolving TAM heterogeneity by mapping cell-types spatially and in their native context has not been reported to the Reviewer's knowledge. Finding the potential location of where polarization takes place (if it does biologically take place) is what spatial mapping can do that no other methods can. This is an important point that I believe is highlighted by the authors, but they may want to think about the messaging since that is certainly novel and provides future investigators a guide for laser capture dissection of this area and downstream proteomics and/or transcriptomics.

Data and methodology: Establishing confidence in antibody performance is important for this manuscript. Especially considering that phenotypic markers are associated with M1 and M2 like functional properties and formalin fixation times of the FFPE blocks likely varied and some of the primary antibodies were RUO. Readers are likely to accept the performance of CD68, CD163, CD206, PDL1 and AE1/AE3. However, IRF8 may raise questions considering the vendor source and Figure 1B appears to show nonspecific staining. Can the authors cite other groups that have previously used this primary antibody? Or perhaps provide the editor with data to support specific staining from the Vendor or internal experiments.

The manuscript as a whole would be of interest to readers of Nature Communications since TAM heterogeneity is a significant problem and existing approaches have not led to a significant understanding of the different roles of TAM. Suggested improvements are provided below.

Major Comments:

1. Abstract

Overall, the abstract is well-written. The authors may want to consider slightly modifying the language to clearly define the challenge that they are focused on. The second sentence states "Whilst TAM heterogeneity in GC is recognized their spatial distributions within tumor regions are not known." The second sentence should highlight why is TAM heterogeneity a problem? What is the significance of TAM heterogeneity? I assume that it is their plasticity and multi-function resulting in problems for immuno-oncology (tumor metastasis, immune suppression and resistance to therapy), however, the authors should clearly state what the challenge is for all readers of Nat Comm. This should follow with what existing methods have done (genomic, single cell level, etc.) to resolve heterogeneity, whether that has helped or not, and what the pitfalls have been. Then introduce the concept that determining the relationship between TAM heterogeneity and spatial organization in the native context may help resolve phenotypic heterogeneity in a meaningful (i.e., functional) way.

Question: In terms of IRF8, CD68, CD163 and CD206, did the authors observe changes or is it more appropriate to state distinct expression profiles were observed according to spatial location?

Question: What was the rationale for selecting IRF8, CD68, CD163 and CD206 as the four TAM markers? The authors may want to consider providing rationale for these markers.

2. Abstract, Line 27-28 "... on TAMs resulted in the TAM population disparity between the tumor site and adjacent normal tissue and between the tumor-nest and stroma..."

Comment: The authors may want to consider changing the phrase "resulted in" to "were associated with" since no mechanistic studies were performed to demonstrate cause and effect relationship.

3. Abstract, Lines 59 "TAM is a diverse population of cells"

4. Introduction, Lines 69-70: "In this study we sought to investigate the TAM spatial distribution landscape and associated PDL1 expression in GC.

Comment: From my perspective, the primary objective of this manuscript was to determine whether spatial organization of TAM can help resolve TAM heterogeneity. A secondary objective was to evaluate the cellular sources of PD-L1 expression. The authors may want to separate these two into two separate sentences since PD-L1 in the current sentence seems to dilute the message.

5. Results, Line 85-86, Figure 1B: "The juxtaposed section was stained with a multiplex IHC panel comprising CD68 (pan-macrophage)..."

Comment: The authors may want to consider changing the word 'juxtaposed section' to 'serially sectioned'

Question: Is CD68 truly a pan-marker for TAMs? If so, the authors may want to cite more papers supporting this claim.

Question: Figure 1B shows that there are many cells that are CD206+ and CD163+ that are CD68 - which further highlights the importance of whether CD68 is a pan-marker of TAMs.

6. Results, Line 88-89, "A total of 1800 high power fields (C:52%, E: 13%, M: 21%, N: 14%) were imaged across all ROIs."

Question: The authors state that a total of 56 whole slide images from 56 patients were analyzed. The number of 20x images per patient can vary widely depending on the surface area of the tissue section. How many 20x images were generated?

7. Results, Figure 1D.

Question: Is the CD68+CD206+ cell shown actually CD68+? The authors may want to consider providing another example (I recognize that finding a single cell with clear double expression may not

be trivial but the assumption that all TAMs are CD68+ is central to the approach in this manuscript).

8. Results, Figure 1F and 1G.

Question: Were the distinct TAM populations between patients found to reside in the same ROI of a tumor?

9. Results, Figure 2B

Question: If the highest data point in the Core group which sits between an M1/M2 Ratio of 4-5 was removed from this set, would there still be significance?

10. Results, Line 130, Figure 2C

Question: Is it correct to state "significant increases" or is it more accurate to state "higher densities"?

11. Results, Line 144 "TAM-composition change is dependent on their proximity to tumor cells"

Question: Is it change dependent? This implies that TAM distances from tumor cells were altered during in vitro experiments with TAM composition being the read out. The authors may want to consider changing this to "TAM-composition is associated with proximity to tumor cells".

12. Results, Line 174-176: "All TAM-tumor pairings peaked within 10-20 μm , suggesting that this distance, which would put cells into direct contact with each other, may be the zone for polarization or phenotypic change of the TAMs in situ"

13. Results, Line 173 "a tumor cells"

Comment: should remove 'a'

14. Results, Figure 5A

Question: Is the PDL1 expression shown here in the Core or the Margin/Edge?

Reviewer #2 (Gastric cancer, tumour microenvironment)(Remarks to the Author):

The paper by Huang and Boussioutas address the spatial resolution of TAM populations in gastric cancer using multiplex IHC techniques. Although TAM heterogeneity has been known, there have been difficulties to spatially organize TAMs based on markers they express. Using the multiplex IHC techniques, the authors successively resolved TAM population disparity in the tumor sites. It was very interesting that the expression patterns of CD206+ TAMs, highly expressed towards stroma, were strongly correlated with PDL1 expression, suggesting new insights for potential therapeutics for gastric cancer. Globally the paper is sound, well written and the conclusions are well supported by results. However, there are a few points that should be improved before accepting the manuscript.

- 1.The author performed great deal of experiments. However, it would be much more useful and informative to give detailed rationale and interpretation of results. It seems that the result section is the complicated lists of what the authors obtained from experiments. The author tried to explain and interpret the data in the discussion section but need more.
- 2.The authors mentioned that the distribution patterns of TAMs at the edge and margin were similar to those of the core (line 154-156). But looking at Figure S3A, patterns of the margin look different from either the core or the edge, the margin being more skewed to the left.
- 3.It is very interesting that patients with CD68++CD163+ TAMs showed improved PFS and OS, if the effective density is considered (line 187-198). However, transcriptomic microarray data from Figure 4B show down-regulated TP53 DNA repair. The authors correlate the high CD68++CD163+ TAM density with an enhanced host immune responses as one of reason behind the improved survival. The authors should provide physical evidence behind this claim or at least provide detailed reasoning.
- 4.PDL1 expression by all TAMs (Figure 5A) is very difficult to visualize. It would be useful to quantify PDL1 expression by all TAMs tested even though the author provided the PDL1 expression of patients in Figure 5B.
- 5.The author claims that the number of PDL1+ macrophages was equivalent to the number of PDL1+ tumor cells (Figure 5D). However, looking at the cell number, it does not quite look equivalent. The percent cell type looks equivalent.
- 6.TAMs exhibited a more M2-like phenotype at the margin compared to a significant increase of M1-like anti-tumor TAMs in the core. CD68+IRF8+ TAMs were more abundant in the core than at the margin and more within the nest than in the stroma. CD68+IRF8+ TAMs are located in the closest proximity to tumor cells, whereas CD68+CD163+CD206+ TAMs were furthest from the tumor cells. However, although the expression of PDL1 in Figure 5B shows the lowest expression level of PDL1 for patients with GC, considering very small cell numbers of CD68+IRF8+ TAMs shown in Figure 5D, CD68+IRF8+ TAMs are expressing relatively significant amount of PDL1. It would be important to explain the relative expression of PDL1 on different TAMs to better understand the significance of the result.
- 7.Since the expression of CD206 in TAMs is closely related to significantly up-regulated expression of PDL1 in M2-like pro-tumoral activity of macrophages, if possible, it would be informative to look at the phosphorylation status of STAT6 or the activity of JAK from the remaining samples.

Reviewer #3 (Gastric cancer)(Remarks to the Author):

This study by Huang et al applies a multiplex immunohistochemistry (IHC) approach to profile subpopulations of tumor-associated macrophages (TAMs) in a cohort of 56 gastric cancers (GCs). Using four macrophage markers (IRF8, CD68, CD163, CD206) and other markers (PDL1, AE1/AE3), the authors classify seven distinct TAM populations. Several of these subgroups are associated with distinct patterns within the tumor, and also distinct patterns of co-existence. They report that increased density of CD68++ CD163 TAMs are associated with improved patient survival, albeit in a manner not independent of other known clinical prognostic factors. In a subset of the GCs, correlation of the TAM populations with microarray based expression profiles revealed associations of specific

expression signatures with TAM populations (eg CD68+ IRF8+ TAMs with Il1 signaling). Finally, they report that CD68+ only, and also CD68+ CD206+ TAMs, exhibit high PDL1 expression, and that in GC TAMs contribute a significant proportion of the total PDL1 intensity (~50%).

Major Comments

1) The markers chosen to apply to the GCs were chosen by the authors. Motivation should be provided regarding the choice of markers, and why other markers might not have been selected. In particular, those related to the classification of macrophages.

2) A key finding of the study involves the defining of seven distinct macrophage subpopulations. The statistical robustness of this finding needs to be demonstrated. For example, does the classification follow a classification hierarchy (eg divide based on CD163 first, then CD206) and if so do the subtypes hold if the ordering of the rules are perturbed. Also, if one does a statistical subsampling of the regions of interest (say 10,000 times), are the seven populations rediscovered with the same expression patterns? Is seven the right number of populations, and not six or eight?

3) Similarly, the statistical robustness of the differences in tumor location with respect to the different tumor regions/sites should be assessed and reported.

4) Is the association with patient survival not observed, if one simply divides the patients based on their bulk TAM number?

5) Do any key results fundamentally change if the core and edge regions are collapsed into one? The distinction seems arbitrary to this Reviewer.

Minor Comments

1) Are the seven populations equally distributed across different GC tumor types and stages?

2) The authors state in the Introduction that the classical interpretation of macrophages as M1 and M2 may be overly simplistic, and that in reality macrophages exist as a continuum of subtypes. In the Discussion, it would be nice if the authors can juxtapose their findings and comment on which findings might have been missed if one were only relying on the M1/M2 classification.

Dear Editor and Reviewers,

Thank you for your considered view, favourable comments and constructive advice, we have incorporated them into our manuscript within word limit. Your comments are addressed in a point-by-point manner as described below.

Our responses were coloured in Blue. Comments were put in the manuscript and also in this document to track our changes. For example: Reviewer #1, Comment 1 will be tagged as R1C1. Figures needed for clarifications or the ones that we felt might not add to our current story in response to the comments were provided as additional pdf files (Figure R1, R2 and R3).

Comment [HY1]: Comments

Reviewers' comments:

Reviewer #1 (Multiplex IHC)(Remarks to the Author):

Reviewer Comments (NCOMMS-18-38415):

Summary: In their manuscript submission, titled "Macrophage Spatial Heterogeneity in Gastric Cancer Defined by Multiplex Immunohistochemistry" the authors highlight the challenge of TAM heterogeneity in GC and sought to determine the relationship between TAM cellular heterogeneity and spatial organization in terms of PD-L1 expression. The authors propose to resolve TAM heterogeneity by spatial mapping of TAM according to four markers – CD68, CD206, CD163 and IRF8.

Originality and interest: Overall, this submission demonstrates significance and originality. This manuscript contributes towards efforts focused on resolving TAM heterogeneity. I think it's important to clarify the language in the abstract and introduction to highlight that resolving TAM heterogeneity by mapping cell-types spatially and in their native context has not been reported to the Reviewer's knowledge. Finding the potential location of where polarization takes place (if it does biologically take place) is what spatial mapping can do that no other methods can. This is an important point that I believe is highlighted by the authors, but they may want to think about the messaging since that is certainly novel and provides future investigators a guide for laser capture dissection of this area and downstream proteomics and/or transcriptomics.

Data and methodology: Establishing confidence in antibody performance is important for this manuscript. Especially considering that phenotypic markers are associated with M1 and M2 like functional properties and formalin fixation times of the FFPE blocks likely varied and some of the primary antibodies were RUO. Readers are likely to accept the performance of CD68, CD163, CD206, PDL1 and AE1/AE3. However, IRF8 may raise questions considering the vendor source and Figure 1B appears to show nonspecific staining. Can the authors cite other groups that have previously used this primary antibody? Or perhaps provide the editor with data to support specific staining from the Vendor or internal experiments.

Larger images of Figure 1B are provided as Figure R1.1. As shown in the image, the IRF8+ non-macrophage (CD68-) signals were not nonspecific staining but true nucleus staining on the other cell types, including tumor cells (AE1/AE3+IRF8+) and some other unknown cell types (CD68-IRF8+; related to Figure S1A). These CD68-IRF8+ cells were abundant within the intra-tumor tertiary lymphoid structures (Figure R1.2A). Healthy donor tonsils which were used as a positive control for optimizing the antibody¹, also showed good quality nucleus staining of IRF8 (Figure R1.2B).

Based on these results, we believe that, at least for this IRF8 antibody from this particular vendor source, is reliable for our IHC approach.

The manuscript as a whole would be of interest to readers of Nature Communications since TAM heterogeneity is a significant problem and existing approaches have not led to a significant understanding of the different roles of TAM. Suggested improvements are provided below.

Major Comments:

1. Abstract

Overall, the abstract is well-written. The authors may want to consider slightly **modifying** the language to clearly define the challenge that they are focused on. The second sentence states “Whilst TAM heterogeneity in GC is recognized their spatial distributions within tumor regions are not known.” The second sentence should highlight why is TAM heterogeneity a problem? What is the significance of TAM

Comment [HY2]: R1C1

heterogeneity? I assume that it is their plasticity and multi-function resulting in problems for immuno-oncology (tumor metastasis, immune suppression and resistance to therapy); however, the authors should clearly state what the challenge is for all readers of Nat Comm. This should follow with what existing methods have done (genomic, single cell level, etc.) to resolve heterogeneity, whether that has helped or not, and what the pitfalls have been. Then introduce the concept that determining the relationship between TAM heterogeneity and spatial organization in the native context may help resolve phenotypic heterogeneity in a meaningful (i.e., functional) way.

Question: In terms of IRF8, CD68, CD163 and CD206, did the authors observe changes or is it more appropriate to state distinct expression profiles were observed according to spatial location?

Comment [HY3]: R1C1

Amended as “Distinct marker expression profiles”.

Question: What was the rationale for selecting IRF8, CD68, CD163 and CD206 as the four TAM markers? The authors may want to consider providing rationale for these markers.

Comment [HY4]: R1C1

Rationale added in the second last paragraph of the Introduction section.

A gating strategy of our multiplex IHC panel is added to Figure 1D.

2. Abstract, Line 27-28 “... on TAMs resulted in the TAM population disparity between the tumor site and adjacent normal tissue and between the tumor-nest and stroma...”

Comment: The authors may want to considering changing the phrase “resulted in” to “were associated with” since no mechanistic studies were performed to demonstrate cause and effect relationship.

Comment [HY5]: R2C2

Amended as suggested.

3. Abstract, Lines 59 “TAM is a diverse population of cells”

Comment [HY6]: R1C3

Amended. “TAMs are a diverse population of cells”

4. Introduction, Lines 69-70: “In this study we sought to investigate the TAM spatial distribution landscape and associated PDL1 expression in GC.

Comment: From my perspective, the primary objective of this manuscript was to determine whether spatial organization of TAM can help resolve TAM heterogeneity. A secondary objective was to evaluate the cellular sources of PD-L1 expression. The authors may want to separate these two into two separate sentences since PD-L1 in the current sentence seems to dilute the message.

Comment [HY7]: R1C4

Amended.

5. Results, Line 85-86, Figure 1B: “The juxtaposed section was stained with a multiplex IHC panel comprising CD68 (pan-macrophage)...”

Comment: The authors may want to consider changing the word ‘juxtaposed section’ to ‘serially sectioned’

Comment [HY8]: R1C5

Amended as suggested.

Question: Is CD68 truly a pan-marker for TAMs? If so, the authors may want to cite more papers supporting this claim.

Comment [HY9]: R1C5

Question: Figure 1B shows that there are many cells that are CD206+ and CD163+ that are CD68 – which further highlights the importance of whether CD68 is a pan-marker of TAMs.

Comment [HY10]: R1C5

Have cited another paper.

We also have a short discussion about the specificity of CD68 in the Discussion section:

“Whilst CD68 is a recognized mononuclear phagocyte marker, low CD68 expression can be found on other cell types. As a result, the CD68-only population is likely to contain a proportion of non-macrophage cells.”

CD68 is currently the best pan-macrophage marker for humans in use. However, it is definitely true that CD68 is not an exclusive marker only for macrophages. Another way to improve the specificity of CD68 is to add a broader lineage marker (e.g., CD11b) to the panel, but due to technological limitations of the number of fluorophores currently available, we selected CD68 as a pan-macrophage marker.

6. Results, Line 88-89, “A total of 1800 high power fields (C:52%, E: 13%, M: 21%, N:

14%) were imaged across all ROIs.”

Question: The authors state that a total of 56 whole slide images from 56 patients were analyzed. The number of 20x images per patient can vary widely depending on the surface area of the tissue section. How many 20x images were generated?

Comment [HY11]: R1C6

The whole slide scans were performed at 10X, and each image was used for selecting ROIs (20X) for each patient.

The total number of images per ROI:

The percentage (C:52%, E: 13%, M: 21%, N: 14%) were rounded from (C: 51.53% , E: 12.65%, M: 21.41%, N: 14.42%)

- Core $1800 \times 51.53\% = 929$
- Edge $1800 \times 12.65\% = 228$
- Margin $1800 \times 21.41\% = 386$
- Normal $1800 \times 14.42\% = 260$

The number of images varied between samples due to the size of the tissue, but all available area were obtained within each sample. The results were normalized using the area of tissue (cell/mm^2 ; Figure 2C).

7. Results, Figure 1D.

Question: Is the CD68+CD206+ cell shown actually CD68+? The authors may want to consider providing another example (I recognize that finding a single cell with clear double expression may not be trivial but the assumption that all TAMs are CD68+ is central to the approach in this manuscript).

Comment [HY12]: R1C7

A new CD68+CD206+ TAM representative image was added (Figure 1E and Figure S1A).

8. Results, Figure 1F and 1G.

Question: Were the distinct TAM populations between patients found to reside in the same ROI of a tumor?

Comment [HY13]: R1C8

Yes. Between different regions of interest, the CD68+IRF8+, CD163+ (CD206-) and CD68+CD206+ TAMs were found more in the Core comparing to the normal gastric tissue (Figure 2C). Between different patient samples, the CD68+IRF8+, CD68++CD163+ and CD6+CD206+ macrophages were found to be colocalized within a same tumor Core (Figure 4A).

Figure 1F and 1G were used for visualization of different TAM population based on their marker expression intensities. Actual location and distribution of these populations were further investigated in Figure 2 and 3.

9. Results, Figure 2B

Question: If the highest data point in the Core group which sits between an M1/M2 Ratio of 4-5 was removed from this set, would there still be significance?

Comment [HY14]: R1C9

Yes, it was still significant (Figure R1.3).

The point with the highest M1/M2 ratio was a microsatellite unstable (MSI) molecular subtype of GC. The high ratio is likely due to the influence of other immune cell types (e.g., T cells).

Note: The p value between Core and Margin is 0.0013, which should have been labelled as ** not *. We have amended this.

10. Results, Line 130, Figure 2C

Question: Is it correct to state “significant increases” or is it more accurate to state “higher densities”?

Comment [HY15]: R1C10

Amended as suggested.

11. Results, Line 144 “TAM-composition change is dependent on their proximity to tumor cells”

Question: Is it change ... dependent? This implies that TAM distances from tumor cells were altered during in vitro experiments with TAM composition being the read out. The authors may want to consider changing this to “TAM-composition is associated with proximity to tumor cells”.

Comment [HY16]: R1C11

Amended as suggested.

12. Results, Line 174-176: "All TAM-tumor pairings peaked within 10-20 μ m, suggesting that this distance, which would put cells into direct contact with each other, may be the zone for polarization or phenotypic change of the TAMs in situ"

Comment [HY17]: R1C12

Amended.

All TAM-tumor pairings peaked within 10-20 μ m, a distance which would put cells into direct contact with each other.

This suggests that this distance, may be the zone for TAM polarization or phenotypic change in situ, and could potentially be used as a threshold for tissue extraction for further genomic/proteomic analyses.

13. Results, Line 173 "a tumor cells"

Comment [HY18]: R1C13

Comment: should remove 'a'

Removed.

14. Results, Figure 5A

Comment [HY19]: R1C14

Question: Is the PDL1 expression shown here in the Core or the Margin/Edge?

In the Core. Added to the Figure legend.

Reference

1. Qi CF, Li Z, Raffeld M, Wang H, Kovalchuk AL, Morse HC, 3rd. Differential expression of IRF8 in subsets of macrophages and dendritic cells and effects of IRF8 deficiency on splenic B cell and macrophage compartments. Immunol Res. 2009;45: 62-74.

Reviewers' comments:

Reviewer #2 (Gastric cancer, tumour microenvironment)(Remarks to the Author):

The paper by Huang and Boussioutas address the spatial resolution of TAM populations in gastric cancer using multiplex IHC techniques. Although TAM heterogeneity has been known, there have been difficulties to spatially organize TAMs based on markers they express. Using the multiplex IHC techniques, the authors successively resolved TAM population disparity in the tumor sites. It was very interesting that the expression patterns of CD206+ TAMs, highly expressed towards stroma, were strongly correlated with PDL1 expression, suggesting new insights for potential therapeutics for gastric cancer. Globally the paper is sound, well written and the conclusions are well supported by results. However, there are a few points that should be improved before accepting the manuscript.

1.The author performed great deal of experiments. However, it would be much more useful and informative to give detailed rationale and interpretation of results. It seems that the result section is the complicated lists of what the authors obtained from experiments. The author tried to explain and interpret the data in the discussion section but need more.

Comment [HY20]: R2C1

Have added more explanation in the Result and Discussion section.

2.The authors mentioned that the distribution patterns of TAMs at the edge and margin were similar to those of the core (line 154-156). But looking at Figure S3A, patterns of the margin look different from either the core or the edge, the margin being more skewed to the left.

Comment [HY21]: R2C2

The images from the previous Figure S3A were representative images from a patient. We have amended this panel to the results of all patient available for the Edge (n=30) and Margin (n=26) (Figure S4A).

The distribution patterns and distance of TAM populations to tumor cells at the Edge was very similar to the Core. At the Margin, although the distance of TAMs to tumor cells were much further (as mentioned by the reviewer), the distribution patterns remain similar to the Core. The CD68+IRF8+ macrophages were the closest to the tumor cells, followed by the CD68+CD206+ and the CD68++CD163+ TAMs. The CD68+CD163+CD206+ macrophages were the furthest away from the tumor cells

but was not different to the CD68+ and CD68+CD206++ TAMs (Figure 3B and Figure S4A).

Based on these results, although the absolute distance was different which might be related to different degree of infiltration at different ROIs, the distribution patterns of TAMs to tumor cells were similar between the Core, Edge and the Margin.

3. It is very interesting that patients with CD68++CD163+ TAMs showed improved PFS and OS, if the effective density is considered (line 187-198). However, transcriptomic microarray data from Figure 4B show down-regulated TP53 DNA repair. The authors correlate the high CD68++CD163+ TAM density with an enhanced host immune responses as one of reason behind the improved survival. The authors should provide physical evidence behind this claim or at least provide detailed reasoning.

Comment [HY22]: R2C3

We have stained the same patient samples with CD3 and CD8 and found the CD3+CD8+ cytotoxic T cells, which has been shown to possess anti-tumor activities in multiple cancer models¹⁻³ [REDACTED], was strongly colocalized within the CD68++CD163+ TAM within the same tumor microenvironment (Figure S6G). This result coincides with our gene-signature analysis that the high immune cell signalling might be the reason behind improved patient survival.

4. PDL1 expression by all TAMs (Figure 5A) is very difficult to visualize. It would be useful to quantify PDL1 expression by all TAMs tested even though the author provided the PDL1 expression of patients in Figure 5B.

Comment [HY23]: R2C4

Figure 5B was the result of quantifying the PDL1 expression by all TAM population but was summarized to a defined number per patient using the mean intensity of each TAM population. This approach is similar to current approach used clinically to score PDL1 positivity staining based on one arbitrary threshold without considering the heterogeneity of PDL1 a TAM population (Clinical approach: >1% of PDL1+ cells in an image, without cell specific characterizations).

We further take this heterogeneity into consideration for the panels within Figure 5 and show that it will be more informative to not only defining PDL1 expression with an arbitrary threshold but to analyze the cells with different intensities (Discussion).

5.The author claims that the number of PDL1+ macrophages was equivalent to the number of PDL1+ tumor cells (Figure 5D). However, looking at the cell number, it does not quite look equivalent. The percent cell type looks equivalent.

Comment [HY24]: R2C5

The difference in number between the PDL1+ tumor cells and macrophages was around 3000 cells (Figure R2.1). We have added pie charts showing the percentage of the two in Figure 5D to visualize this data interpretation. The percentage of PDL1+ tumor cells was 50.79% and for all PDL1+ macrophages was 49.21%, which was very similar.

To avoid confusion, we have amended the “equivalent” in the text to “similar”.

6.TAMs exhibited a more M2-like phenotype at the margin compared to a significant increase of M1-like anti-tumor TAMs in the core. CD68+IRF8+ TAMs were more abundant in the core than at the margin and more within the nest than in the stroma. CD68+IRF8+ TAMs are located in the closest proximity to tumor cells, whereas CD68+CD163+CD206+ TAMs were furthest from the tumor cells. However, although the expression of PDL1 in Figure 5B shows the lowest expression level of PDL1 for patients with GC, considering very small cell numbers of CD68+IRF8+ TAMs shown in Figure 5D, CD68+IRF8+ TAMs are expressing relatively significant amount of PDL1. It would be important to explain the relative expression of PDL1 on different TAMs to better understand the significance of the result.

Comment [HY25]: R2C6

Agreed, have added to the Discussion.

7.Since the expression of CD206 in TAMs is closely related to significantly up-regulated expression of PDL1 in M2-like pro-tumoral activity of macrophages, if possible, it would be informative to look at the phosphorylation status of STAT6 or the activity of JAK from the remaining samples.

Comment [HY26]: R2C7

We performed experiments in light of the reviewer comments. [REDACTED]

Results:

[REDACTED]

Conclusions and Discussions:

[REDACTED]

We have performed the experiments and have provided the results but have not included this data in the manuscript given the limitation of text and figures.

Reference

1. Wang Y, Zhu C, Song W, Li J, Zhao G, Cao H. PD-L1 Expression and CD8(+) T Cell Infiltration Predict a Favorable Prognosis in Advanced Gastric Cancer. *J Immunol Res.* 2018;2018: 4180517.
2. Ziai J, Gilbert HN, Foreman O, et al. CD8+ T cell infiltration in breast and colon cancer: A histologic and statistical analysis. *PLoS One.* 2018;13: e0190158.

3. Ye SL, Li XY, Zhao K, Feng T. High expression of CD8 predicts favorable prognosis in patients with lung adenocarcinoma: A cohort study. *Medicine (Baltimore)*. 2017;96: e6472.
4. Tomita K, Caramori G, Ito K, et al. STAT6 expression in T cells, alveolar macrophages and bronchial biopsies of normal and asthmatic subjects. *J Inflamm (Lond)*. 2012;9: 5.
5. Wick KR, Berton MT. IL-4 induces serine phosphorylation of the STAT6 transactivation domain in B lymphocytes. *Mol Immunol*. 2000;37: 641-652.
6. Pesu M, Takaluoma K, Aittomaki S, et al. Interleukin-4-induced transcriptional activation by stat6 involves multiple serine/threonine kinase pathways and serine phosphorylation of stat6. *Blood*. 2000;95: 494-502.

Reviewers' comments:

Reviewer #3 (Gastric cancer)(Remarks to the Author):

This study by Huang et al applies a multiplex immunohistochemistry (IHC) approach to profile subpopulations of tumor-associated macrophages (TAMs) in a cohort of 56 gastric cancers (GCs). Using four macrophage markers (IRF8, CD68, CD163, CD206) and other markers (PDL1, AE1/AE3), the authors classify seven distinct TAM populations. Several of these subgroups are associated with distinct patterns within the tumor, and also distinct patterns of co-existence. They report that increased density of CD68⁺⁺ CD163 TAMs are associated with improved patient survival, albeit in a manner not independent of other known clinical prognostic factors. In a subset of the GCs, correlation of the TAM populations with microarray based expression profiles revealed associations of specific expression signatures with TAM populations (eg CD68⁺ IRF8⁺ TAMs with IL1 signaling). Finally, they report that CD68⁺ only, and also CD68⁺ CD206⁺ TAMs, exhibit high PDL1 expression, and that in GC TAMs contribute a significant proportion of the total PDL1 intensity (~50%).

Major Comments

1) The markers chosen to apply to the GCs were chosen by the authors. Motivation should be provided regarding the choice of markers, and why other markers might not have been selected. In particular, those related to the classification of macrophages.

Comment [HY27]: R3C1

Have included the rationale into Introduction, and a panel for marker gating strategy (Figure 1D).

2) A key finding of the study involves the defining of seven distinct macrophage subpopulations. The statistical robustness of this finding needs to be demonstrated.

For example, does the classification follow a classification hierarchy (eg divide based on CD163 first, then CD206) and if so do the subtypes hold of the ordering of the rules are perturbed.

Comment [HY28]: R3C2

Also, if one does a statistical subsampling of the regions of interest (say 10,000 times), are the seven populations rediscovered with the same expression patterns? Is seven the right number of populations, and not six or eight?

Comment [HY29]: R3C2

3) Similarly, the statistical robustness of the differences in tumor location with respect to the different tumor regions/sites should be assessed and reported.

Comment [HY30]: R3C3

*The responses below are for both R3C2 and R3C3. The results were incorporated in the manuscript as **Figure S2** and in the **Method** section.*

Our multiplex IHC panel was designed based on Flow cytometry gating strategy using the markers and current knowledge of macrophage polarization^{1, 2}. We expected that all macrophages were CD68 positive; the M2-like macrophage were CD163 and CD206 positive, and the M1-like were IRF8 positive (CD163 and CD206 negative; Figure 1D). The subgrouping of TAMs into seven predominant populations was done using this gating strategy with a supervised software³⁻⁶ (inForm, Method).

(A)

To test the robustness of our TAM subgrouping approach, we subsampled 100 cells from each of the seven TAM populations per patient within each ROI (Core, Edge, and Margin). The subsampling was repeated for 2000 times (Figure S2A).

The results suggest that for each patient subsampled data (Figure S2A), although the expression levels various between patient tissues, the TAM populations still follow the same grouping strategies (Figure 1D). To test the statistical robustness of the subsampling, we compared the median marker expression signatures of the subsample cohort (each TAM population in each patient per ROI was compared) to the signature of the overall cohort (Figure 1F) using the Pearson correlation ($p < 0.05$).

Our results show that in each patient in each ROI, the accuracy of the subsampling of each TAM population was above 99% in the Core and Edge and at least 96% accurate at the Margin (Figure S2A), which reflects the overall robustness of our TAM subgrouping approach using the supervised inForm software.

(B)

To validate the number of populations, we performed unsupervised clustering method on the same subsampled cohort and computed the “within group sum of squares (wss)” for different ROIs using the K-means clustering method^{7, 8} (Figure S2B). Based on the “elbow method”^{7, 8}, the results suggest that for all ROIs, the optimal number of clusters was around 5 to 7.

To test if the groupings using unsupervised clustering methods were similar our supervised approach, we compared the median marker expression signatures from each clusters with the expression signature of the TAM populations (Figure 1F) to

validate if our TAM populations can be rediscovered (Pearson, $p < 0.01$; Figure S2C and Figure R3.1A).

The results indicate that within the optimal clustering range (5-7), all TAM populations can be rediscovered, and were much better comparing to 2-4 numbers of clustering (Figure S2C).

However, due to the limited number of markers can be used for multiplex IHC, unsupervised clustering method cannot differentiate the difference between the CD68⁺⁺CD163⁺/CD68⁺CD163⁺ pair and the CD68⁺CD206⁺⁺/CD68⁺CD206⁺ pair (Figure S2D). This explains why seven populations can be identified when only 5-6 clusters. However, using the gating strategy and supervised methods, we showed that there were clearly differences between these populations.

The median CD68 expression of CD68⁺⁺CD163⁺ about 2-folds higher than the CD68⁺CD163⁺ TAMs and there's a 1.5-fold difference in the CD206 expression between the 2 CD206⁺ (CD163⁻) populations (Figure 1F-H, Figure S1C-G and Figure R3.1C).

More importantly, we showed that these populations have visible, and statically spatial location preferences (Figure 2, Figure 3, Figure 7, Figure S1D-E, and Figure R3.1B), which has not been taken into account when using unsupervised clustering only based on marker expression. It would be expected that a population of TAM has location preference, even if it was in small number, is likely that they are biologically relevant. Therefore, we believe that the distribution of TAMs should be taken into account for identifying TAM populations, which is often ignored using unsupervised clustering methods just based on expression of markers.

(C)

Because our panel was designed based on current literature, this will put the CD163 and CD206 as the first selection criteria down the hierarchy (Figure 1D). However, using our current results, we cannot conclude if there is a classification hierarchy based on CD163 or CD206 (Figure R3.1A).

As we can see in clustering number 3, the discovery of CD163⁺ (CD206⁻) and CD206⁺ (CD163⁻) were inconsistent and sometimes exclusive between different subsampling cohorts across all ROIs. This indicates that there's no classification hierarchy for these two markers in our current panel, given that the CD68⁺CD163⁺CD206⁺ was the only population that was overlapping between the

two markers and it does not matter which of the CD163 or CD206 were used first this population can be discovered.

Although our result does not support there's a classification hierarchy between the CD163 and CD206, we cannot conclude that in reality there isn't. CD163 is highly upregulated in the IL-10 polarized macrophages⁹. CD206 is more widely expressed in all M2-like macrophages (IL-4/IL-10)¹⁰, tissue macrophages², and also unpolarized naïve macrophages¹⁰. We believe that more markers, as used in CyToF and flow cytometry, will be needed to investigate the priority of different markers.

Conclusions:

1. We demonstrated the robustness of TAM groupings using our supervised clustering methods that the accuracies were above 96% across all ROIs.
2. We cannot conclude the priority of marker using the multiplex IHC panel due to panel design and the limitation of current technology.
3. "Seven" is within the optimal number of TAM population that can be clustered using unsupervised/supervised clustering method with our current panel.

However, we do not emphasize that 7 is an absolute. Rather, we state "7 predominant populations distributed within a nonlinear spectrum". Our data, as well as others¹¹⁻¹⁴, clearly showed the macrophage population is a continuum and the number of populations will likely change if more markers can be added to our staining panel. Although the distribution of the additional markers on current groupings of TAM will need to be assessed individually, the change we observe with our current panel between different ROIs (decrease of CD206 and increase of CD163, CD68, IRF8 from the Margin into the Core; Figure 7) will likely remain, and could possibly be used as a basis for further TAM population mapping in GC.

4) Is the association with patient survival not observed, if one simply divides the patients based on their bulk TAM number?

Comment [HY31]: R3C4

We did not see association with patient survival using bulk TAM number (Figure S5A). We saw survival association using the M1/M2 classification that the M2 macrophage (CD163 and/or CD206 positive) was a predictor of improved patient survival (Figure S5B). This signal was coming from the CD68++CD163+ macrophages (Figure S3E-F) and not from other M2-like TAMs (Figure S5C-D).

5) Do any key results fundamentally change if the core and edge regions are collapsed into one? The distinction seems arbitrary to this Reviewer.

Comment [HY32]: R3C5

There were no differences if the Core and Edge regions were collapsed. The Edge was merely around 1mm from the Margin into the tumor site, whereas the Core was the major body of the tumor mass. However, although we did not observe differences in macrophage populations between the two regions, we found higher T cells at the Edge comparing to the Core (Wang et al. submitted). This result suggests that there is a difference between the Edge and Core in terms of the balance between the adaptive and innate immunity, which might be biologically relevant.

In addition, for some patients that we have both Affymetrix data and FFPE blocks (multiplex IHC) available, tumor Core region was not available for some due to sampling issue. In order to be more confident about the data correlation between the two platforms, including Edge could potentially increase the accuracy of the results (Figure 4B).

Minor Comments

1) Are the seven populations equally distributed across different GC tumor types and stages?

Comment [HY33]: R3MC1

There was a trend of lower TAM cell densities in the Diffuse GC comparing to the Intestinal GC (Figure S3). However, the densities of tumor cells and other non-macrophage cells were also lower in the Diffuse GC due to the nature that the Diffuse GC tends to invade deeper into adjacent normal tissue therefore these samples will likely contain more stromal component. We do not have enough evidence to conclude whether this decrease TAM densities in Diffuse GC was due to sampling issue or is really biological relevant (Figure S3D). This will be a consideration for future experiments

No differences were observed between the four molecular GC subtypes (Figure S3E).

No significance or trend of TAM distribution difference was observed between different stages (Figure S3F).

2) The authors state in the Introduction that the classical interpretation of macrophages as M1 and M2 may be overly simplistic, and that in reality macrophages exist as a continuum of subtypes. In the Discussion, it would be nice if

the authors can juxtapose their findings and comment on which findings might have been missed if one were only relying on the M1/M2 classification.

Comment [HY34]: R3MC2

Thank you, have added to the Discussion section.

References

1. Norton SE, Dunn ET, McCall JL, Munro F, Kemp RA. Gut macrophage phenotype is dependent on the tumor microenvironment in colorectal cancer. *Clin Transl Immunology*. 2016;5: e76.
2. Roszer T. Understanding the Mysterious M2 Macrophage through Activation Markers and Effector Mechanisms. *Mediators Inflamm*. 2015;2015: 816460.
3. Stack EC, Wang C, Roman KA, Hoyt CC. Multiplexed immunohistochemistry, imaging, and quantitation: a review, with an assessment of Tyramide signal amplification, multispectral imaging and multiplex analysis. *Methods*. 2014;70: 46-58.
4. Halse H, Colebatch AJ, Petrone P, et al. Multiplex immunohistochemistry accurately defines the immune context of metastatic melanoma. *Sci Rep*. 2018;8: 11158.
5. Carstens JL, Correa de Sampaio P, Yang D, et al. Spatial computation of intratumoral T cells correlates with survival of patients with pancreatic cancer. *Nat Commun*. 2017;8: 15095.
6. Gorris MAJ, Halilovic A, Rabold K, et al. Eight-Color Multiplex Immunohistochemistry for Simultaneous Detection of Multiple Immune Checkpoint Molecules within the Tumor Microenvironment. *J Immunol*. 2018;200: 347-354.
7. Cancer Genome Atlas Research N. Comprehensive molecular characterization of gastric adenocarcinoma. *Nature*. 2014;513: 202-209.
8. Kakushadze Z, Yu W. *K-means and cluster models for cancer signatures. *Biomol Detect Quantif*. 2017;13: 7-31.
9. Lurier EB, Dalton D, Dampier W, et al. Transcriptome analysis of IL-10-stimulated (M2c) macrophages by next-generation sequencing. *Immunobiology*. 2017;222: 847-856.
10. Roussel M, Ferrell PB, Jr., Greenplate AR, et al. Mass cytometry deep phenotyping of human mononuclear phagocytes and myeloid-derived suppressor cells from human blood and bone marrow. *J Leukoc Biol*. 2017;102: 437-447.
11. Lavin Y, Kobayashi S, Leader A, et al. Innate Immune Landscape in Early Lung Adenocarcinoma by Paired Single-Cell Analyses. *Cell*. 2017;169: 750-765 e717.

12. Tsujikawa T, Kumar S, Borkar RN, et al. Quantitative Multiplex Immunohistochemistry Reveals Myeloid-Inflamed Tumor-Immune Complexity Associated with Poor Prognosis. *Cell Rep.* 2017;19: 203-217.
13. Chevrier S, Levine JH, Zanutelli VRT, et al. An Immune Atlas of Clear Cell Renal Cell Carcinoma. *Cell.* 2017;169: 736-749 e718.
14. Xue J, Schmidt SV, Sander J, et al. Transcriptome-based network analysis reveals a spectrum model of human macrophage activation. *Immunity.* 2014;40: 274-288.

Figure R1.1

Figure R1.2

Figure R1.3

Patient#	1	2	3	4	5	6	7	8	9	10	11	12
Ratio	4.477134	2.286395	2.252903	1.994423	1.651969	1.645567	1.463742	1.320243	1.296619	1.25716	1.232046	1.085446
Patient#	13	14	15	16	17	18	19	20	21	22	23	24
Ratio	1.060916	1.038615	0.98519	0.948644	0.870686	0.862445	0.848718	0.834203	0.825979	0.812389	0.795065	0.786438
Patient#	25	26	27	28	29	30	31	32	33	34	35	36
Ratio	0.731619	0.718532	0.715182	0.710499	0.667481	0.636461	0.623808	0.58318	0.580215	0.549872	0.544424	0.530681
Patient#	37	38	39	40	41	42	43	44	45	46		
Ratio	0.483385	0.439677	0.356876	0.316537	0.296046	0.251227	0.208807	0.17055	0.071422	0.046965		

Figure R2.1

		PDL1+ cell type	1	2	3	4	5
Number	AE1AE3		88601	88689	88702	88572	88314
	All Macrophage		85851	85927	85735	85521	85731
	Total cell		174452	174616	174437	174093	174045
Percentage	AE1AE3		50.79%	50.79%	50.85%	50.88%	50.74%
	All Macrophage		49.21%	49.21%	49.15%	49.12%	49.26%

Figure R3.1

C Expression signature - Pearson

Expression signature - Pearson

REVIEWERS' COMMENTS:

Reviewer #1 (Remarks to the Author):

Reviewer Comments (NCOMMS-18-38415A):

Summary: In their revised manuscript submission, titled "Macrophage Spatial Heterogeneity in Gastric Cancer Defined by Multiplex Immunohistochemistry" the authors present the problem of tumor-associated macrophage (TAM) heterogeneity in gastric cancer (GC) and sought to determine the relationship between spatial-based macrophage cellular organization and macrophage heterogeneity.

Overall, this revised submission the authors have adequately addressed the reviewer comments. Specifically, the authors seek to help resolve TAM heterogeneity based on macrophage populations in their native spatial context. This manuscript demonstrates significance and originality (assuming there are no other significant papers that have been previously published examining this relationship in gastric cancer – I am not aware of any). It is the opinion of this Reviewer, that this manuscript would be of interest to readers of Nature Communications since TAM heterogeneity is a significant clinical problem and existing approaches have not translated to clinical understanding of the different roles of TAM.

Minor comments to the authors responses and revisions are provided below.

Minor Comments:

1. Abstract, Line 2: "To understand the influences of tumor contexts on TAMs heterogeneity, we performed multiplex immunohistochemistry on human GC cases and identified seven predominant TAM populations based on combinations of markers."

Comment: The authors should consider changing "influences of tumor contexts on TAMs heterogeneity" to "relationship between tumor context and TAM heterogeneity" since the term "influences" implies a causal relationship.

2. Line 3: "Distinct marker expression profiles on TAMs were associated with their population disparity between the tumor site and adjacent normal tissue and between the tumor-nest and stroma."

Comment: "Population disparity" in this sentence is unclear. Does the following sentence communicate the same concept? "Distinct TAM marker expression profiles were found at the interface of tumor and adjacent normal tissue when compared to the interface of tumor-nest and stroma."

3. Results, Lines 139-142: "Higher densities observed for the CD68+CD163+, CD68++CD163+ and CD68+CD206+ macrophages within the tumor regions comparing to the normal tissue suggested that these populations were been polarized locally within the tumor microenvironment."

Comment: The authors should consider changing this sentence for grammatical reasons: "Higher densities of CD68+CD163+, CD68++CD163+ and CD68+CD206+ macrophage cells populations were found in distinct tumor regions when compared to normal tissues suggesting that these TAM populations were polarized according to location in the tumor microenvironment."

4. Results, Lines 186-188: "Collectively, our data showed that the predominant TAM populations differ

phenotypically between (i) tumor and adjacent normal tissue (Figure 2) and (ii) when in close proximity to the tumor cells (Figure 3B-D)."

Comment: Shouldn't this sentence be in present tense? And perhaps clearer in structure. "Collectively, our data shows that the predominant TAM population differs phenotypically in the interface zone of tumor and adjacent normal tissue (Figure 2) when compared the TAM population in close proximity to the tumor cells (Figure 3B-D)."

5. Results, Lines 202–206: "The results showed that patients with an increased effective density of CD68++CD163+ macrophages had a much better prognosis with PFS and also OS (Figure 3F and Figure S4D) comparing to the patients with lower TAM density."

Comment: The word "increased" suggests the effective density was measured at two different time points in the same tumor. Also, the phrase "much better" is not scientifically meaningful to readers.

Perhaps the authors should consider stating "The results showed that patients with a higher effective density of CD68++CD163+ macrophages were associated with significantly longer PFS and OS (Figure 3F and Figure S4D) when compared to patients with lower TAM density."

Methods, Line 644-645: "Acquired images (1800 in total) were analyzed..."

Comment: The authors should change this part of the sentence to "Acquired images (n=1800) were analyzed..."

Reviewer #3 (Remarks to the Author):

The Authors have responded adequately to my original concerns. Assuming the manuscript advances further in the publication process, I would suggest the authors address the following minor changes:

1) A section should be included in the Main Text describing how they have assessed the statistical robustness of clusters. In this current version, this is not explicitly stated in the Results.

2) The revised statements in the main text, while factually accurate, are riddled with grammatical errors that need to be corrected prior to publication.

REVIEWERS' COMMENTS:

Reviewer #1 (Remarks to the Author):

Reviewer Comments (NCOMMS-18-38415A):

Summary: In their revised manuscript submission, titled “Macrophage Spatial Heterogeneity in Gastric Cancer Defined by Multiplex Immunohistochemistry” the authors present the problem of tumor-associated macrophage (TAM) heterogeneity in gastric cancer (GC) and sought to determine the relationship between spatial-based macrophage cellular organization and macrophage heterogeneity.

Overall, this revised submission the authors have adequately addressed the reviewer comments. Specifically, the authors seek to help resolve TAM heterogeneity based on macrophage populations in their native spatial context. This manuscript demonstrates significance and originality (assuming there are no other significant papers that have been previously published examining this relationship in gastric cancer – I am not aware of any). It is the opinion of this Reviewer, that this manuscript would be of interest to readers of Nature Communications since TAM heterogeneity is a significant clinical problem and existing approaches have not translated to clinical understanding of the different roles of TAM.

Minor comments to the authors responses and revisions are provided below.

Minor Comments:

1. Abstract, Line 2: “To understand the influences of tumor contexts on TAMs heterogeneity, we performed multiplex immunohistochemistry on human GC cases and identified seven predominant TAM populations based on combinations of markers.”

Comment: The authors should consider changing “influences of tumor contexts on TAMs heterogeneity” to “relationship between tumor context and TAM heterogeneity” since the term “influences” implies a causal relationship.

Amended as suggested.

2. Line 3: “Distinct marker expression profiles on TAMs were associated with their population disparity between the tumor site and adjacent normal tissue and between

the tumor-nest and stroma.”

Comment: “Population disparity” in this sentence is unclear. Does the following sentence communicate the same concept? “Distinct TAM marker expression profiles were found at the interface of tumor and adjacent normal tissue when compared to the interface of tumor-nest and stroma.”

This part of the Abstract was amended as:

“Using distinct expression marker profiles on TAMs, we report seven predominant populations distributed between tumor and non-tumor tissue.”

3. Results, Lines 139-142: “Higher densities observed for the CD68+CD163+, CD68++CD163+ and CD68+CD206+ macrophages within the tumor regions comparing to the normal tissue suggested that these populations were been polarized locally within the tumor microenvironment.”

Comment: The authors should consider changing this sentence for grammatical reasons: “Higher densities of CD68+CD163+, CD68++CD163+ and CD68+CD206+ macrophage cells populations were found in distinct tumor regions when compared to normal tissues suggesting that these TAM populations were polarized according to location in the tumor microenvironment.”

Amended as “Higher densities of CD68+CD163+, CD68++CD163+ and CD68+CD206+ macrophages were found within the tumor regions when compared to normal tissues suggesting that these populations were polarized according to their location in the tumor microenvironment”.

4. Results, Lines 186-188: “Collectively, our data showed that the predominant TAM populations differ phenotypically between (i) tumor and adjacent normal tissue (Figure 2) and (ii) when in close proximity to the tumor cells (Figure 3B-D).”

Comment: Shouldn't this sentence be in present tense? And perhaps clearer in structure. “Collectively, our data shows that the predominant TAM population differs phenotypically in the interface zone of tumor and adjacent normal tissue (Figure 2) when compared the TAM population in close proximity to the tumor cells (Figure 3B-D).”

Amended as “Collectively, our data shows that the predominant TAM population

differs phenotypically between the tumor and adjacent normal tissue (Fig. 2). The same phenotypic differences were observed in macrophages proximal to the tumor cells (Fig. 3b-d).

5. Results, Lines 202–206: “The results showed that patients with an increased effective density of CD68++CD163+ macrophages had a much better prognosis with PFS and also OS (Figure 3F and Figure S4D) comparing to the patients with lower TAM density.”

Comment: The word “increased” suggests the effective density was measured at two different time points in the same tumor. Also, the phrase “much better” is not scientifically meaningful to readers.

Perhaps the authors should consider stating “The results showed that patients with a higher effective density of CD68++CD163+ macrophages were associated with significantly longer PFS and OS (Figure 3F and Figure S4D) when compared to patients with lower TAM density.”

Amended as “The results showed that patients with a higher effective density of CD68++CD163+ macrophages had significantly longer RFS and OS (Fig. 3f and Supplementary Fig. 4d) when compared to patients with lower TAM density.”

Methods, Line 644-645: “Acquired images (1800 in total) were analyzed...”

Comment: The authors should change this part of the sentence to “Acquired images (n=1800) were analyzed...”

Amended as suggested. Line

Reviewer #3 (Remarks to the Author):

The Authors have responded adequately to my original concerns. Assuming the manuscript advances further in the publication process, I would suggest the authors address the following minor changes:

1) A section should be included in the Main Text describing how they have assessed the statistical robustness of clusters. In this current version, this is not explicitly stated in the Results.

Due to the limitation of word count we have not describe the validation in detail in the Main text. The information is put in the Method and also in the Supplementary Fig. 2 with the validation results.

2) The revised statements in the main text, while factually accurate, are riddled with grammatical errors that need to be corrected prior to publication.

Language modified.